# Dual Caption Preference Optimization for Diffusion Models

**Amir Saeidi**[*]                                                                                    *ssaeidi1@asu.edu*
*School of Computing and Augmented Intelligence*
*Arizona State University*

**Yiran Lawrence Luo**[*]                                                                              *yluo97@asu.edu*
*School of Computing and Augmented Intelligence*
*Arizona State University*

**Agneet Chatterjee**                                                                                 *agneet@asu.edu*
*School of Computing and Augmented Intelligence*
*Arizona State University*

**Shamanthak Hegde**                                                                                  *shegde23@asu.edu*
*School of Computing and Augmented Intelligence*
*Arizona State University*

**Bimsara Pathiraja**                                                                                 *bpathir1@asu.edu*
*School of Computing and Augmented Intelligence*
*Arizona State University*

**Yezhou Yang**                                                                                       *yz.yang@asu.edu*
*School of Computing and Augmented Intelligence*
*Arizona State University*

**Chitta Baral**                                                                                      *chitta@asu.edu*
*School of Computing and Augmented Intelligence*
*Arizona State University*

**Reviewed on OpenReview:** *https://openreview.net/forum?id=ruZksIJBBd*

## Abstract

Recent advancements in human preference optimization, originally developed for Large Language Models (LLMs), have shown significant potential in improving text-to-image diffusion models. These methods aim to learn the distribution of preferred samples while distinguishing them from less preferred ones. However, within the existing preference datasets, the original caption often does not clearly favor the preferred image over the alternative, which weakens the supervision signal available during training. To address this issue, we introduce **Dual Caption Preference Optimization (DCPO)**, a data augmentation and optimization framework that reinforces the learning signal by assigning two distinct captions to each preference pair. This encourages the model to better differentiate between preferred and less-preferred outcomes during training. We also construct **Pick-Double Caption**, a modified version of Pick-a-Pic v2 with separate captions for each image, and propose three different strategies for generating distinct captions: captioning, perturbation, and hybrid methods. Our experiments show that DCPO significantly improves image quality and relevance to prompts, outperforming Stable Diffusion (SD) 2.1, SFT$_{\text{Chosen}}$, Diffusion-DPO and MaPO across multiple metrics, including Pickscore, HPSv2.1, GenEval, CLIPscore, and ImageReward, fine-tuned on SD 2.1 as the backbone.

---

[*]Equal contribution. Correspondence to ssaeidi1@asu.edu

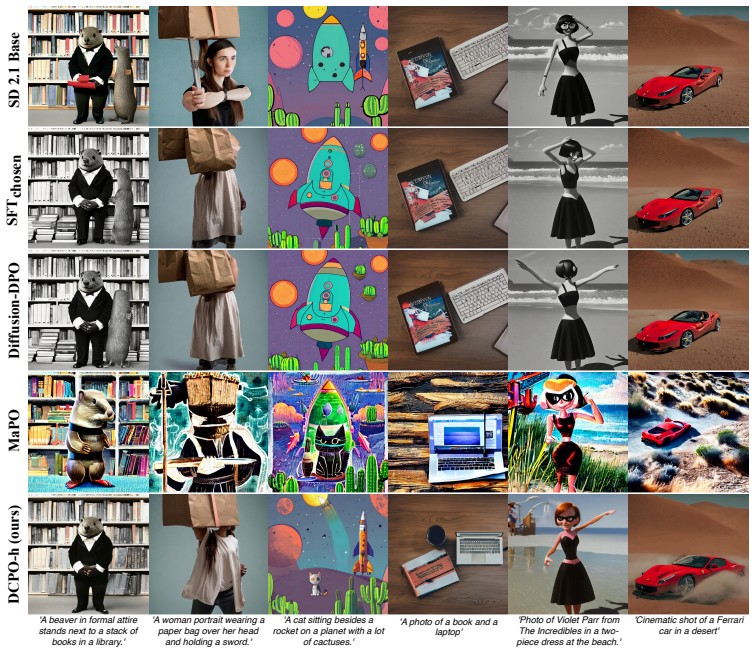

Figure 1: Sample images generated by different methods on the HPSv2, Geneval, and Pickscore benchmarks. After fine-tuning SD 2.1 with SFT$_{\text{Chosen}}$, Diffusion-DPO, MaPO, and DCPO on Pick-a-Picv2 and Pick-Double Caption datasets, DCPO produces images with notably higher preference and visual appeal (See more examples in Appendix H).

# 1 Introduction

Image synthesis models (Rombach et al., 2022; Esser et al., 2024) have achieved remarkable advancements in generating photo-realistic and high-quality images. Text-conditioned diffusion (Song et al., 2020) models have led this progress due to their strong generalization abilities and proficiency in modeling high-dimensional data distributions. As a result, they have found wide range of applications in image editing (Brooks et al., 2023), video generation (Wu et al., 2023a) and robotics (Carvalho et al., 2023). Consequently, efforts have focused on aligning them with human preferences, targeting specific attributes like safety (Liu et al., 2024b), style (Everaert et al., 2023), and personalization (Ruiz et al., 2023), thereby improving their usability and adaptability.

Similar to the alignment process of Large Language Models (LLMs), aligning diffusion models involves two main steps: **1.** Pre-training and **2.** Supervised Fine-Tuning (SFT). Recent fine-tuning based methods have been introduced to optimize diffusion models according to human preferences by leveraging Reinforcement Learning with Human Feedback (RLHF) (Ouyang et al., 2022), the aim of which is to maximize an explicit reward. However, challenges such as fine-tuning a separate reward model and reward hacking have led to the adoption of Direct Preference Optimization (DPO) (Rafailov et al., 2024) techniques like Diffusion-DPO (Wallace et al., 2024). Intuitively, Diffusion-DPO involves maximizing the difference between a preferred image and a less preferred image for a given prompt.

Although DPO-based methods are effective in comparison to SFT-based approaches, applying direct optimization in multi-modal settings presents certain challenges. Current preference optimization datasets consist of a preferred ($x^w$) and a less preferred ($x^l$) image for a given prompt ($c$). Ideally, $x^w$ should show a higher correlation with $c$ compared to $x^l$. However, we find that in current datasets, the preferred and less-preferred images exhibit a substantial overlap in their semantic distributions for the given prompt $c$, which we refer to as *semantic overlap* in the data. Additionally, irrelevant information in prompt $c$ restricts the U-Net's ability to predict noises from $x^l$ in the diffusion reverse process, which we refer to as *irrelevant prompts*. This entails that there is a lack of sufficient distinguishing features between the two pairs ($x^w$, $c$), ($x^l$, $c$), thereby increasing the complexity of the optimization process.

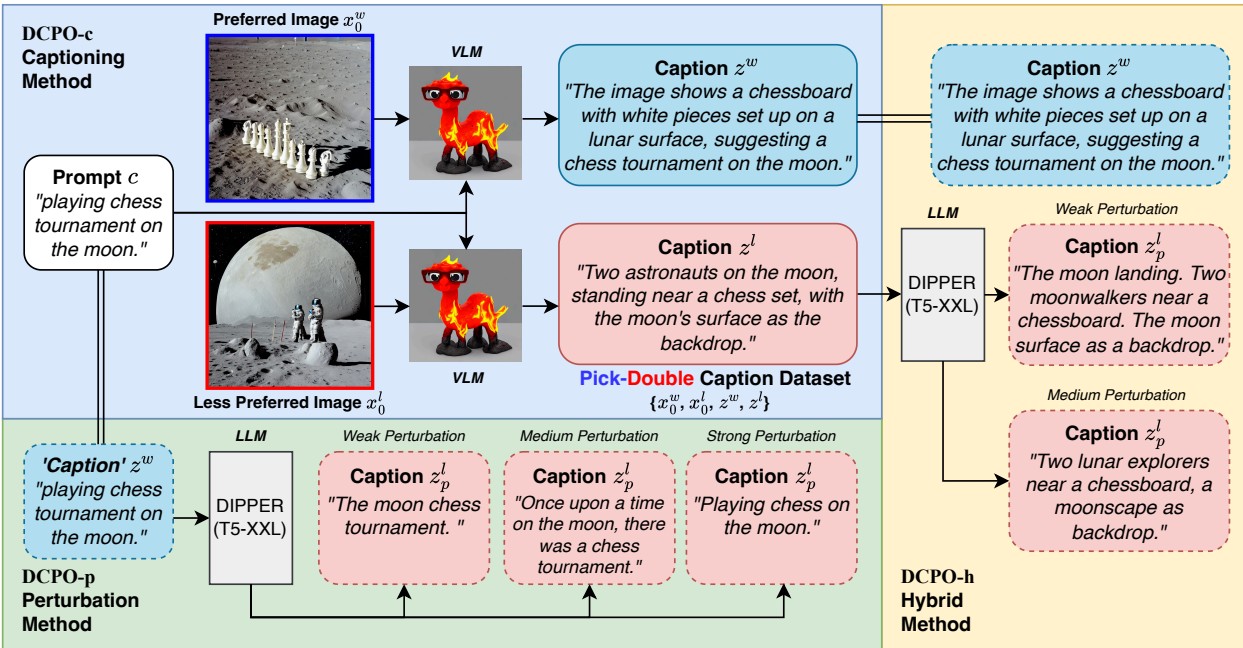

Figure 2: The DCPO pipeline in 3 variants: DCPO-c, DCPO-p, and DCPO-h, all of which require a duo of a captioned preferred image $(x_0^w, z^w)$ and a captioned less-preferred image $(x_0^l, z^l)$. **DCPO-c (Top Left):** We use a captioning model to generate distinctive captions respectively for images $x_0^w$ and $x_0^l$ given the shared prompt $c$. **DCPO-p (Bottom Left):** We take prompt $c$ as the caption for image $x_0^w$, then we use a Large Language Model (LLM) to generate a semantically perturbed prompt $z_p^l$ given prompt $c$ as the caption for image $x_0^l$. **DCPO-h (Right):** A hybrid method where the generated caption $z^l$ is now perturbed into $z_p^l$ for image $x_0^l$. Our *Pick-Double Caption* Dataset discussed in Section 4.1 is constructed using DCPO-c.

To address the aforementioned bottleneck, we propose **DCPO: Dual Caption Preference Optimization**, a novel preference optimization technique designed to align diffusion models by utilizing two distinct captions corresponding to the preferred and less preferred image. DCPO broadly consists of two steps: a text generation framework that develops better-aligned captions and a novel objective function that utilizes these captions as part of the training process.

The text generation framework aims to mitigate the *semantic overlap* between the distributions of preferred and less-preferred images, given the same prompts, in existing datasets. We hypothesize that prompt $c$ does not serve as the optimal signal for optimization because they do not convey the reasons why an image is preferred or dis-preferred; based on the above, we devise the following techniques to generate better aligned captions. The *first* method involves using a captioning model $Q_\phi(z^i|x^i, c)$; which generates a new prompt $z^i$ based on an image $x^i$ and the original prompt $c$, where $i \in (w, l)$. The *second* method introduces perturbation techniques $f$, such that $c = z^w, z^l = f(c)$; i.e., generating $z^l$, to represent the less preferred image, considering the original prompt $c$ as the prompt aligned with the preferred image. We investigate multiple semantic variants of $f$, where each variant differs in the degree of perturbation applied to the original caption $c$. Finally, we also explore a hybrid combination of the above methods, where we combine the strong prior of the captioning model and the efficient nature of the perturbation method. All the above methods are designed to generate captions that effectively discriminate between the preferred and less preferred images.

We introduce a novel objective function that allows DCPO to incorporate $z^w$ and $z^l$ into its optimization process. Specifically, during optimization, the policy model $p_\theta$ increases the likelihood of the preferred image $x^w$ conditioned on the prompt $z^w$, while simultaneously decreasing the likelihood of the less preferred image $x^l$ conditioned on the prompt $z^l$. The results in Tables 1 and 2 demonstrate that DCPO consistently outperforms other methods, with notable improvements of +0.21 in Pickscore, +0.31 in HPSv2.1, +1.8 in

normalized ImageReward, +0.15 in CLIPscore, and +2% in GenEval. Additionally, DCPO achieved 58% in general preference and 66% in visual appeal compared to Diffusion-DPO on the PartiPrompts dataset, as evaluated by GPT-4o (see Figure 7).

In summary, our contributions are as follows :

- **Double Caption Generation**: We introduce the Captioning and Perturbation methods to address the issue of overlapping semantic distributions in current datasets, as illustrated in Figure 3. In the Captioning method, we employ state-of-the-art models like LLaVA (Liu et al., 2024a) and Emu2 (Sun et al., 2024) to generate a caption $z$ based on the image $x$ and prompt $c$. Additionally, we use DIPPER (Krishna et al., 2024), a paraphrase generation model built by fine-tuning the T5-XXL model to create three levels of perturbation from the prompt $c$.

- **Dual Caption Preference Optimization (DCPO):** We propose DCPO, a modified version of Diffusion-DPO, that leverages the U-Net encoder embedding space for preference optimization. This method enhances diffusion models by aligning them more closely with human preferences, using two distinct captions for the preferred and less preferred images during optimization.

- **Improved Model Performance:** We demonstrate that our approach significantly outperforms SD 2.1, SFT, Diffusion-DPO, and MaPO across metrics such as Pickscore, HPSv2.1, GenEval, CLIPscore, normalized ImageReward, and GPT-4o (Achiam et al., 2023) evaluations.

## 2 Preliminary

Aligning a generative model typically involves fine-tuning it to produce outputs that are more aligned with human preferences. Estimating the reward model $r$ based on human preference is generally challenging, as we do not have direct access to the reward model. However, if we assume the availability of ranked data generated under a given condition $c$, where $x_0^w \succ x_0^l | c$ (with $x_0^w$ representing the preferred sample and $x_0^l$ the less-preferred sample), we can apply the Bradley-Terry theory to model these preferences. The Bradley-Terry (BT) model expresses human preferences as follows:

$$p_{BT}(x_0^w \succ x_0^l | c) = \sigma(r(c, x_0^w) - r(c, x_0^l)) \tag{1}$$

where $\sigma(\cdot)$ is the sigmoid function, and $r(x_0, c)$ is derived from a neural network parameterized by $\phi$.

Subsequently, $\phi$ is estimated by maximum likelihood training for binary classification as follows:

$$L_{BT}(\phi) = -\mathbb{E}_{c, x_0^w, x_0^l}[\log \sigma(r_\phi(c, x_0^w) - r_\phi(c, x_0^l))] \tag{2}$$

where prompt $c$ and data pair $(x_0^w, x_0^l)$ are sourced from a human-annotated dataset.

This approach to reward modeling has gained popularity in aligning large language models, particularly when combined with reinforcement learning (RL) techniques like proximal policy optimization (PPO) (Schulman et al., 2017) to fine-tune the model based on rewards learned from human preferences, known as Reinforcement Learning from Human Feedback (RLHF) (Ouyang et al., 2022). The goal of RLHF is to optimize the conditional distribution $p(x_0|c)$ (where $c \sim D_c$) such that the reward model $r(c, x_0)$ is maximized, while keeping the policy model within the desired distribution using a KL-divergence term to ensure it remains reachable under the following objective:

$$\max_{p_\theta} \mathbb{E}_{c \sim \mathcal{D}_c, x_0 \sim p_\theta(x_0|c)}[r(c, x_0)] - \beta \mathbb{D}_{KL}[p_\theta(x_0|c)||p_{\text{ref}}(x_0|c)] \tag{3}$$

where $\beta$ controls how far the policy model $p_\theta$ can deviate from the reference model $p_{ref}$.

It can be demonstrated that the objective in Equation 3 converges to the following policy model:

$$p_\theta^*(x_0|c) = p_{\text{ref}}(x_0|c) \exp(r(c, x_0)/\beta)/Z(c) \tag{4}$$

where $Z$ is the partition function.

The training objective for $p_\theta$, inspired by DPO, has been derived to be equivalent to Equation 4 without the need for an explicit reward model $r(x, c)$. Instead, RLHF learns directly from the preference data $(c, x_0^w, x_0^l) \sim D$:

$$L_{\text{DPO}}(\theta) = -\mathbb{E}_{c, x_0^w, x_0^l}\left[ \log \sigma\left( \beta \log \frac{p_\theta(x_0^w|c)}{p_{\text{ref}}(x_0^w|c)} - \beta \log \frac{p_\theta(x_0^l|c)}{p_{\text{ref}}(x_0^l|c)} \right)\right] \tag{5}$$

where $\sigma(\cdot)$ is the sigmoid function.

Through this re-parameterization, instead of optimizing the reward function $r$ before applying reinforcement learning, the RLHF method directly optimizes the conditional distribution $p_\theta(x_0|c)$.

## 3 Method

In this section, we present the *semantic overlap* issue preference datasets, where the semantic distribution of preferred and less-preferred images generated from the same prompt $c$ exhibit significant similarity. We also explain the *irrelevant prompt* issue found in previous direct preference optimization methods. To address these challenges, we propose **Dual Caption Preference Optimization (DCPO)**, a method that uses distinct captions for preferred and less preferred images to improve diffusion model alignment.

### 3.1 The Challenges

Generally, to optimize a Large Language Model (LLM) using preference algorithms, we need a dataset $D = \{c, y^w, y^l\}$, where $y^w$ and $y^l$ represent the preferred and less preferred responses to a given prompt $c$. Recent studies (Sun et al., 2025; Deng et al., 2025) have shown that high semantic similarity between preferred and less-preferred responses negatively impacts preference optimization and should be carefully considered during dataset curation. Similarly, in diffusion model alignment, the semantic distributions of preferred and less preferred images should be distinct for the same prompt $c$. However, our analysis shows a substantial overlap between these distributions, which we call *semantic overlap*, as illustrated in Figure 3. For more details, refer to Appendix B.1.

Another issue emerges when direct preference optimizes a diffusion model. In the reverse denoising process, the U-Net model predicts noise for both preferred and less preferred images using the same prompt $c$. As prompt $c$ is more relevant to the preferred image, it becomes less effective for predicting the less preferred one, leading to reduced performance. We call this the *irrelevant prompts* problem.

### 3.2 DCPO: Dual Caption Preference Optimization

Motivated by the *semantic overlap* and *irrelevant prompts* issues, we propose DCPO, a new preference optimization method that optimizes diffusion models using two distinct captions. DCPO is a refined version of Diffusion-DPO designed to address these challenges. More details are in Appendix A.

We start with a fixed dataset $D = \{c, x_0^w, x_0^l\}$, where each entry contains a prompt $c$ and a pair of images generated by a reference model $p_{ref}$. The human labels indicate a preference,

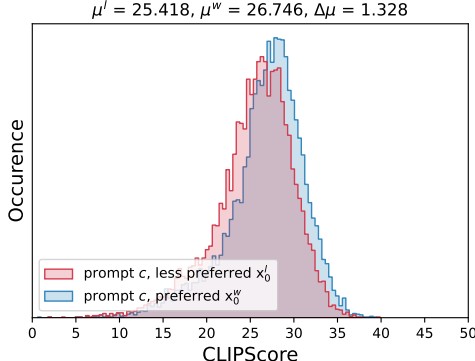

Figure 3: The *semantic overlap* issue in the Pick-a-Pic v2 dataset. $\mu^l$ and $\mu^w$ represent the average CLIPscore of preferred and less preferred images for prompt $c$, respectively. Also, $\Delta\mu$ shows the difference between the distributions.

with $x_0^w$ preferred over $x_0^l$. We assume the existence of a model $R_\phi(z|c,x)$, which generates a caption $z$ given a prompt $c$ and an image $x$. Using this model, we transform the dataset into $D' = \{z^w, z^l, x_0^w, x_0^l\}$, where $z^w$ and $z^l$ are captions for the preferred image $x_0^w$ and the less-preferred image $x_0^l$, respectively. Our goal is to train a new model $p_\theta$, aligned with human preferences, to generate outputs that are more desirable than those produced by the reference model.

The objective of RLHF is to maximize the reward $r(c, x_0)$ for the reverse process $p_\theta(x_{0:T}|z)$, while maintaining alignment with the original reference reverse process distribution. Building on prior work (Wallace et al., 2024), the DCPO objective is defined by direct optimization through the conditional distribution $p_\theta(x_{0:T}|z)$ as follows:

$$\mathcal{L}_{\text{DCPO}}(\theta) = -\mathbb{E}_{(x_0^w, x_0^l) \sim \mathcal{D}'} \log \sigma \Big[ \beta \mathbb{E}_{x_{1:T}^w \sim p_\theta(x_{1:T}^w|x_0^w, z^w), x_{1:T}^l \sim p_\theta(x_{1:T}^l|x_0^l, z^l)} \Big( \log \frac{p_\theta(x_{0:T}^w|z^w)}{p_{\text{ref}}(x_{0:T}^w|z^w)} - \log \frac{p_\theta(x_{0:T}^l|z^l)}{p_{\text{ref}}(x_{0:T}^l|z^l)} \Big) \Big] \tag{6}$$

where $\log[\cdot]$ is the sigmoid function.

However, as noted in Diffusion-DPO (Wallace et al., 2024), the sampling process $x_{1:T} \sim p(x_{1:T}|x_0)$ is inefficient and intractable. To overcome this, we follow a similar approach by applying Jensen's inequality and utilizing the convexity of the $-\log(\cdot)$ function to bring the expectation outside. By approximating the reverse process $p_\theta(x_{1:T}|x_0, z)$ with the forward process $q(x_{1:T}|x_0)$, and through algebraic manipulation and simplification, the DCPO loss can be expressed as:

$$\mathcal{L}_{\text{DCPO}}(\theta) = -\mathbb{E}_{(x_0^w, x_0^l) \sim \mathcal{D}', t \sim \mu(0,T), x_t^w \sim q(x_t^w|x_0^w), x_t^l \sim q(x_t^l|x_0^l)}$$
$$\log \sigma \Big[ -\beta T w(\lambda_t) \Big( (||\epsilon^w - \epsilon_\theta(x_t^w, z^w, t)||_2^2 - ||\epsilon^w - \epsilon_{\text{ref}}(x_t^w, z^w, t)||_2^2) - (||\epsilon^l - \epsilon_\theta(x_t^l, z^l, t)||_2^2 - ||\epsilon^l - \epsilon_{\text{ref}}(x_t^l, z^l, t)||_2^2) \Big) \Big] \tag{7}$$

where $x_t^* = \alpha_t x_0^* + \sigma_t \epsilon^*$, and $\epsilon^* \sim \mathcal{N}(0, I)$ is a sample drawn from $q(x_t^*|x_0^*)$. $\lambda_t = \alpha_t^2/\sigma_t^2$ represents the signal-to-noise ratio, and $\omega(\lambda_t)$ is a weighting function.

To optimize a diffusion model using DCPO, a dataset $D = \{z^w, z^l, x_0^w, x_0^l\}$ is required, where captions are paired with the images. However, the current preference dataset only contains prompts $c$ and image pairs without captions. To address this, we propose three methods for generating captions $z$ and introduce a new high-quality dataset, *Pick-Double Caption*, which provides specific captions for each image, based on Pick-a-Pic v2 (Kirstain et al., 2023).

### 3.2.1 DCPO-c: Captioning Method

In this method, the captioning model $Q_\phi(z|c,x)$ generates the caption $z$ based on the image $x$ and the original prompt $c$. As a result, we obtain a preferred caption $z^w \sim Q_\phi(z^w|c, x^w)$ for the preferred image and a less preferred caption $z^l \sim Q_\phi(z^l|c, x^l)$ for the less preferred image, as illustrated in a sample in Figure 2. Thus, based on the generated captions $z^w$ and $z^l$, we can optimize a diffusion model using the DCPO method.

In the experiment section, we evaluate the performance of DCPO-c and demonstrate that this method effectively mitigates the *semantic overlap* by creating two differentiable distributions. However, the question of how much divergence is needed between the two distributions remains. To investigate this, we propose Hypothesis 1.

**Hyphothesis 1.** *Let $d(z, x)$ represent the semantic distribution between a caption $z$ and an image $x$, with $\mu$ being the mean of the distribution $d$, and $\Delta\mu = \mu(d(z_0^w, x_0^w)) - \mu(d(z_0^l, x_0^l))$ as the difference between the two distributions. Increasing $\Delta\mu$ between the preferred and less-preferred image distributions in a preference dataset beyond a threshold $t$ (i.e., $\Delta\mu > t$), can improve the performance of the model $p_\theta$.*

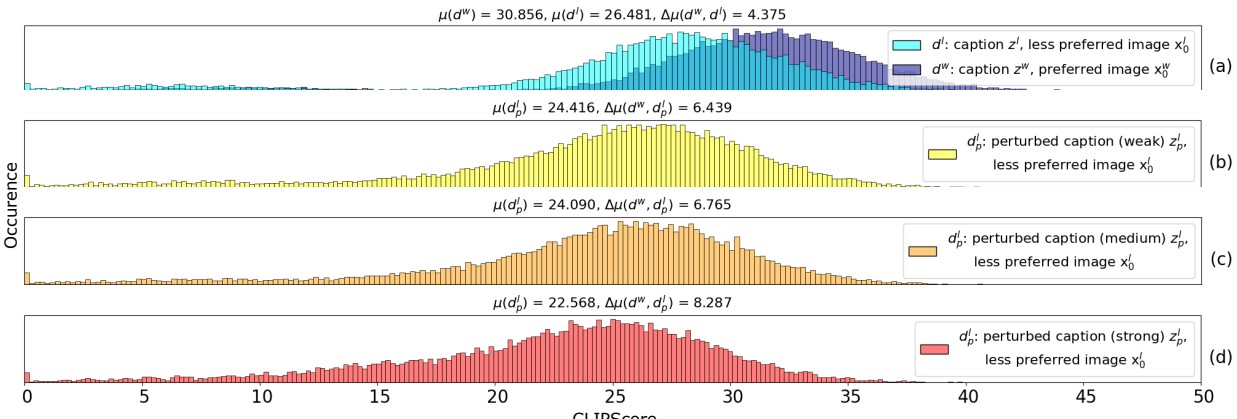

Figure 4: Effect of the perturbation method on semantic distributions in terms of CLIPScore. **(a)** shows the distributions that feature the captions $z^w$ and $z^l$ generated by the LLaVA model, while **(b)**, **(c)**, and **(d)** represent different levels of perturbation on top of the caption $z^l$. The figure demonstrates that as the level of perturbation increases, the distance between the distributions of captions $z^w$ and $z^l$ increases. For more details on the perturbation method, refer to Appendix D.

Our hypothesis suggests that increasing the distance between the two distributions up to a certain threshold $t$ can improve alignment performance. To examine this, we propose the perturbation method to control the distance between the two distributions, represented by $\Delta\mu$.

### 3.2.2 DCPO-p: Perturbation Method

While using a captioning model is an effective way to address the *semantic overlap*, it risks deviating from the original distribution of prompt $c$, and the distributions of preferred and less preferred images may still remain close. To tackle these issues, we propose a perturbation method. In this approach, we assume that prompt $c$ is highly relevant to the preferred image $x_0^w$ and aim to generate a less relevant caption, denoted as $c_p$, based on prompt $c$. To achieve this, we use the model $W_\phi(c_p|c)$, which generates a perturbed version of prompt $c$, altering its semantic meaning. In this framework, prompt $c$ corresponds to the preferred caption $z^w$ ($c = z^w$), while the perturbed prompt $c_p$ represents the less-preferred caption $z^l$ ($c_p = z^l$).

For the perturbation model $W_\phi$, we choose the highly customizeable DIPPER model (Krishna et al., 2024), which is built by fine-tuning T5-XXL (Chung et al., 2022), to produce a degraded version of the prompt $c$. DIPPER is equipped with flexible steerability for controllable textual perturbation, which supports adjustable hyperparameters that control both lexical diversity and word order variation at the output.

We define three levels of perturbation strength: **1) Weak:** where prompt $c_p$ has high semantic similarity to prompt $c$, with minimal differences. **2) Medium:** where the semantic difference between prompt $c_p$ and $c$ is more pronounced than in the weak level. **3) Strong:** where the majority of the semantics in prompt $c_p$ differ significantly from prompt $c$.

**Perturbation Selection Process.** Given each input prompt $c$, we first perform a hyperparameter sweep over DIPPER's controllable dimensions to construct a larger diverse pool of perturbed prompt candidates. We then select the candidates that are well-separated along the CLIPScore scale from the pool, ensuring a clear semantic gradient while always preserving the original input as the one with the highest image-text alignment in terms of CLIPScore affinity. The three resulting perturbed prompts $c_p$ — in levels of **Weak**, **Medium**, and **Strong**, respectively — reflect progressively degraded semantic fidelity relative to the associated image of the input, visualized as the shifting distributions in terms of CLIPScore semantic similarity in Figure 4. For further details on the perturbation method, see Appendix D.

Compared with DCPO-c, DCPO-p is, first of all, less computationally costly. As described in Figure 2, DCPO-p requires only the textual perturbation as the less-preferred caption. In addition, by using the original prompt $c$ directly as the preferred caption, DCPO-p keeps the original distribution at bay, mitigating

DCPO-c's risk where both the preferred and the less preferred may deviate together. However, we observe that the quality of the captions in DCPO-c outperforms that of the original prompt $c$, as shown in Table 9 in Appendix C. Based on this observation, we propose a hybrid method to improve the alignment performance by combining the captioning and perturbation techniques from DCPO-c and DCPO-p, altogether.

### 3.2.3 DCPO-h: Hybrid Method

In this method, instead of perturbing the prompt $c$, we perturb the caption $z$ generated by the model $Q_\phi(z|x,c)$ based on the image $x$ and prompt $c$. As discussed in Section 3.2.1, the goal of the perturbation method is to increase the distance between the two distributions. However, the correlation between the image $x_0$ and prompt $c$ significantly impacts alignment performance. Therefore, we propose Hypothesis 2.

**Hypothesis 2.** *Let $S(c,x)$ represent the correlation score between prompt $c$ and image $x$, and $P(p_\theta(c_1, c_2))$ denote the performance of model $p_\theta$ optimized on captions $c_1$ and $c_2$ with DCPO, where $W_\phi$ is the perturbation model. If $S(z,x) > S(c,x)$, then $P(p_\theta(z^w, z_p^w \sim W_\phi(z_p^w|z^w))) > P(p_\theta(c, c_p \sim W_\phi(c_p|c)))$.*

In Section 4.3, we provide experimental evidence supporting Hypothesis 2 and investigate the potential of using $z_p^l \sim W_\phi(z_p^l|z^l)$ as the less-preferred caption $z^l$, instead of $z_p^w \sim W_\phi(z_p^w|z^w)$ as originally proposed in Hypothesis 2.

## 4 Experiments

We fine-tuned the U-Net model of Stable Diffusion (SD) 2.1 using DCPO on the *Pick-Double Caption* dataset and compared it with SD 2.1 models fine-tuned with $SFT_{Chosen}$, Diffusion-DPO, and MaPO on Pick-a-Picv2 in various metrics. We first describe the *Pick-Double Caption* dataset and compare it to Pick-a-Picv2. Subsequently, we provide an in-depth analysis of the results. Details on fine-tuning are in Appendix F, and further comparisons are in Appendix B.

Table 1: Results on PickScore, HPSv2.1, ImageReward (normalized), and CLIPScore. We show that DCPO significantly improves on Pickscore, HPSv2.1, and ImageReward.

| | Pick Score (↑) | HPSv2.1 (↑) | Image Reward (↑) | CLIP Score (↑) |
|---|---|---|---|---|
| *Results from other methods* | | | | |
| SD 2.1 | 20.30 | 25.17 | 55.8 | 26.84 |
| $SFT_{Chosen}$ | 20.35 | 25.09 | 56.4 | 26.98 |
| Diffusion-DPO | 20.36 | 25.10 | 56.4 | 26.98 |
| MaPO | 20.31 | 25.31 | 55.6 | 26.78 |
| *Results from our methods* | | | | |
| DCPO-c (LLaVA) | 20.46 | 25.10 | 56.5 | 27.00 |
| DCPO-c (Emu2) | 20.46 | 25.06 | 56.6 | 26.97 |
| DCPO-p | 20.28 | 25.42 | 54.2 | 26.98 |
| DCPO-h (LLaVA) | **20.57** | **25.62** | **58.2** | **27.13** |

### 4.1 Pick-Double Caption Dataset

Motivated by the *semantic overlap* observed in previous preference datasets, we applied the captioning method described in Section 3.2.1 to generate unique captions for each image in the Pick-a-Pic v2 dataset. For the *Pick-Double Caption* dataset, we sampled 20,000 instances from Pick-a-Pic v2 and cleaned the samples as detailed in Appendix C. We then employed two state-of-the-art captioning models, LLaVa-1.6-34B and Emu2-37B, to generate captions for both the preferred and less preferred images, as shown in Figure 2.

To generate the captions, we used two different prompting strategies: 1) **Conditional prompt:** where the model was explicitly instructed to generate a caption for image $x$ based on the given prompt $c$, and 2) **Non-conditional prompt:** where the model provided a general description of the image in one sentence without referring to a specific prompt. More details are in Appendix C.

We evaluated the captions generated by LLaVA and Emu2 using CLIPscore, which revealed several key insights. LLaVA produced captions that have more correlation with the images for both preferred and less preferred samples compared to Emu2 and the original captions, although LLaVA's captions were significantly longer (see Table 9 in Appendix C). Models fine-tuned on captions from the conditional prompt strategy outperformed those using the non-conditional approach, though the conditional prompt captions were twice as long. Interestingly, despite Emu2 generating much shorter captions, the models fine-tuned on Emu2 were comparable to those fine-tuned on the original prompts from Pick-a-Pic v2.

Table 2: Results on the GenEval Benchmark. DCPO contributes to model performance in generating the correct number of objects, improving image quality in terms of colors, and constructing attributes accurately.

| Method | Overall | Single object | Two objects | Counting | Colors | Position | Attribute binding |
|---|---|---|---|---|---|---|---|
| *Results from other methods* | | | | | | | |
| SD 2.1[2] | 0.4775 | 0.96 | **0.52** | 0.35 | 0.80 | **0.09** | 0.15 |
| SFT$_{\text{Chosen}}$ | 0.4797 | **1.00** | 0.42 | 0.42 | 0.81 | 0.07 | 0.14 |
| Diffusion-DPO | 0.4857 | 0.99 | 0.48 | 0.46 | 0.83 | 0.04 | 0.11 |
| MaPO | 0.4938 | **1.00** | 0.45 | 0.45 | 0.80 | **0.09** | 0.16 |
| *Results from our methods* | | | | | | | |
| DCPO-c (LLaVA) | 0.4971 | **1.00** | 0.43 | 0.53 | **0.85** | 0.02 | 0.14 |
| DCPO-c (Emu2) | 0.4925 | **1.00** | 0.41 | 0.50 | **0.85** | 0.04 | 0.15 |
| DCPO-p | 0.4906 | **1.00** | 0.41 | 0.50 | 0.83 | 0.03 | **0.17** |
| DCPO-h (LLaVA) | **0.5100** | 0.99 | 0.51 | **0.54** | 0.84 | 0.05 | 0.14 |

Table 3: Performance comparison of DCPO-h and DCPO-p across different perturbation levels. The perturbation method has a strong impact on captions that are more closely correlated with images.

| Method | Pair Caption | Perturbed Level | Pickscore ($\uparrow$) | HPSv2.1 ($\uparrow$) | ImageReward ($\uparrow$) | CLIPscore ($\uparrow$) | GenEval ($\uparrow$) |
|---|---|---|---|---|---|---|---|
| DCPO-p | $(c, c_p)$ | weak | 20.28 | 25.42 | 54.20 | 26.98 | 0.4906 |
| DCPO-h | $(z^w, z_p^w)$ | weak | 20.55 | 25.61 | 57.70 | 27.07 | **0.5070** |
| DCPO-h | $(z^w, z_p^l)$ | weak | **20.58** | **25.70** | **58.10** | **27.15** | 0.5060 |
| DCPO-p | $(c, c_p)$ | medium | 20.21 | 25.34 | 53.10 | 26.87 | 0.4852 |
| DCPO-h | $(z^w, z_p^w)$ | medium | 20.59 | **25.73** | **58.47** | 27.12 | 0.5008 |
| DCPO-h | $(z^w, z_p^l)$ | medium | 20.57 | 25.62 | 58.20 | **27.13** | **0.5100** |
| DCPO-p | $(c, c_p)$ | strong | 20.31 | 25.06 | 54.60 | 27.03 | 0.4868 |
| DCPO-h | $(z^w, z_p^w)$ | strong | 20.57 | 25.27 | 57.43 | 27.18 | **0.5110** |
| DCPO-h | $(z^w, z_p^l)$ | strong | **20.58** | **25.43** | **57.90** | **27.21** | 0.4993 |

A key challenge is generating captions for the less preferred images using the captioning method. We observed that in both prompting strategies, the captions for the preferred images are more aligned with the original prompt $c$ distribution. However, the non-conditional prompt strategy often produces captions for less preferred images that are out-of-distribution (OOD) from the original prompt $c$ in most cases. We will explore this further in Section 4.3.

Finally, we observe that the key advantage of the *Pick-Double Caption* dataset is the greater difference in CLIPscore ($\Delta\mu$) between preferred and less preferred images compared to the original prompts. Specifically, while the original prompt has a $\Delta\mu$ of **1.3**, LLaVA shows a much larger difference at **4.3**, and Emu2 at **2.8**. This increased gap reflects improved alignment performance in models fine-tuned on this dataset, indicating that the captioning method mitigates the *semantic overlap*.

## 4.2 Performance Comparisons

As shown in Table 1, we evaluated all methods on 2,500 unique prompts from the Pick-a-Picv2 (Kirstain et al., 2023) dataset, measuring performance using Pickscore (Kirstain et al., 2023), CLIPscore (Hessel et al., 2022), and Normalized ImageReward (Xu et al., 2023). We also generated images from 3,200 prompts in the HPSv2 (Wu et al., 2023b) benchmark and evaluated them using the HPSv2.1 model. To provide a more fine-grained evaluation, in Table 2, we also compared the methods using GenEval (Ghosh et al., 2023), focusing on how well the fine-tuned models generated images with the correct number of objects, accurate colors, and proper object positioning.

We compared our different variants of DCPO, including the captioning (DCPO-c), perturbation (DCPO-p), and hybrid (DCPO-h) methods, with other approaches, as outlined in Section 3.2. For more information on the fine-tuning process of the models, refer to Appendix F.

---

[2]Note that we rerun all the models on the same seeds to have a fair comparison.

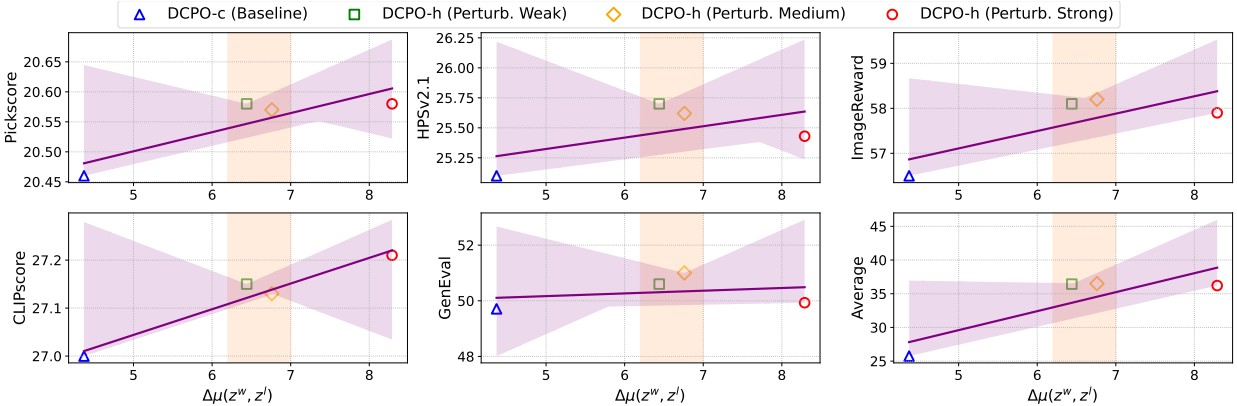

Figure 5: Performance comparison of DCPO-c and DCPO-h on different perturbation levels. We plotted regression lines for the four models, showing that as $\Delta\mu$ increases, performance improves but drops after a threshold $t$ (orange boundary).

The results in Tables 1 and 2 show that DCPO-h significantly outperforms the best scores from other methods, with improvements of $+\mathbf{0.21}$ in Pickscore, $+\mathbf{0.31}$ in HPSv2.1, $+\mathbf{1.8}$ in ImageReward, $+\mathbf{0.15}$ in CLIPscore, and $+\mathbf{2\%}$ in GenEval. Additionally, the results demonstrate that DCPO-c outperforms all other methods on GenEval, Pickscore, and CLIPscore. While DCPO-p performs slightly worse than DCPO-c, it still exceeds SD 2.1, SFT, Diffusion-DPO, and MaPO on GenEval. However, its scores on ImageReward and Pickscore suggest that it underperforms compared to the other approaches. Importantly, DCPO-p shows significant improvement over the other methods on HPSv2.1, highlighting the effectiveness of the perturbation method.

### 4.3 Ablation Studies and Analyses

**Support of Hypothesis 1.** As described in Section 3.2.2, we defined three levels of perturbation: weak, medium, and strong. In Hypothesis 1, we proposed that increasing the distance between the distributions of preferred and less preferred images $\Delta\mu$ improves model alignment performance. To explore this, we fine-tuned SD 2.1 using the DCPO-h method with three levels of perturbation applied to the less preferred captions $z^l$ generated by LLaVA. The results in Figure 5 show that increasing the distance $\Delta\mu$ between the two distributions enhances performance. However, this distance must be controlled and kept below a threshold $t$, a hyperparameter that may vary depending on the task. These findings support our hypothesis.

**Support of Hypothesis 2.** To illustrate the impact of the correlation between the prompt $c$ and image $x$ on the perturbation method, we perturbed both the original prompt $c$ and the less preferred caption $z^w$, generated by the model $Q_\phi$, where $z^w \sim W_\phi(z^w|Q_\phi(z^w|x^w, c))$. At the same time, we kept the caption generated by $Q_\phi$ for the preferred image as the preferred caption, $z^w \sim Q(z^w|x^w, c)$. In this case, we assume $Q_\phi = $ LLaVA and $W_\phi = $ DIPPER. The results in Table 9 in Appendix C show that the caption $z$ generated by LLaVA is more correlated with the image $x$ than the original prompt $c$, indicating that $S(z, x) > S(c, x)$. Based on the results in Table 3, we conclude that perturbing more correlated captions leads to better performance.

**In- vs. Out-of Distribution.** We evaluated DCPO on in-distribution and out-of-distribution (OOD) data. As discussed in Section 4.1, the captioning model can generate OOD captions. To explore this, we fine-tuned SD 2.1 with DCPO-h using LLaVA and Emu2 captions at a medium perturbation level. Figure 6 shows that in-distribution data significantly improve alignment performance, while OOD results for LLaVA in GenEval, Pickscore, and CLIPscore are comparable to Diffusion-DPO. Similar behavior was observed for DCPO-c, as noted in Appendix F.

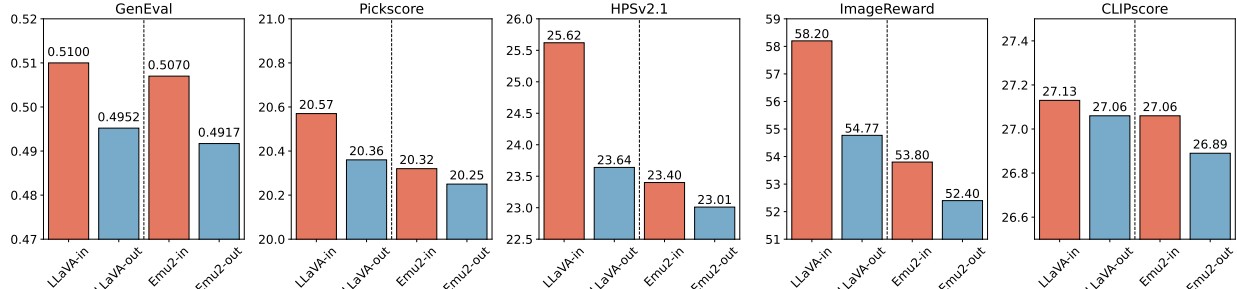

Figure 6: Comparison of DCPO-h performance on in-distribution and out-of-distribution data.

Table 4: Performance comparison of DCPO and Diffusion-DPO fine-tuned on the *Pick-Double Caption* dataset. While larger captions improve the performance of Diffusion-DPO, DCPO-h still significantly outperforms Diffusion-DPO.

| Method | Input Prompt | Token Length (Avg) | Pickscore (↑) | HPSv2.1 (↑) | ImageReward (↑) | CLIPscore (↑) | GenEval (↑) |
|---|---|---|---|---|---|---|---|
| Diffusion-DPO | prompt $c$ | 15.95 | 20.36 | 25.10 | 56.4 | 26.98 | 0.4857 |
| Diffusion-DPO | caption $z^w$ (LLaVA) | 32.32 | 20.40 | 25.19 | 56.6 | 27.10 | 0.4958 |
| Diffusion-DPO | caption $z^w$ (Emu2) | 7.75 | 20.36 | 25.08 | 56.3 | 26.98 | 0.4960 |
| DCPO-h (LLaVA) | Pair $(z^w, z^l_p)$ | (32.32, 31.17) | **20.57** | **25.62** | **58.2** | 27.13 | 0.5100 |
| DCPO-h (LLaVA) | Pair $(z^w, z^w_p)$ | (32.32, 27.01) | **20.57** | 25.27 | 57.4 | **27.18** | **0.5110** |

**Effectiveness of the DCPO.** Our analysis shows that LLaVA captions are twice the length of the original prompt $c$, raising the question of *whether DCPO's improvement is due to data quality or the optimization method.* To explore this, we fine-tuned SD 2.1 with Diffusion-DPO using LLaVA and Emu2 captions instead of the original prompt. The results in Table 4 show that models fine-tuned on LLaVA captions outperform Diffusion-DPO with the original prompt. However, DCPO-h still surpasses the new Diffusion-DPO models, demonstrating the effectiveness of the proposed optimization algorithm.

**Hyperparameter Tuning.** We conducted a systematic hyperparameter study for a fair comparison across DCPO and all baselines. Under a controlled protocol with batch size fixed at 128 and learning rate fixed at $(1 \times 10^{-8})$, we swept $\beta$ for DCPO and Diffusion-DPO over the set $[500, 1000, 2000, 2500, 5000]$. DCPO achieved its strongest performance at a $\beta$ value of 5000, outperforming alternatives on Pickscore, GenEval, HPSv2.1, ImageReward, and CLIPscore, while Diffusion-DPO reached its optimum at a $\beta$ value of 2000. SFT performed best with batch size 64 and learning rate $(1 \times 10^{-9})$. MaPO attained its optimum at a much smaller coefficient, with a $\beta$ value of 0.01, indicating a preference for lighter regularization relative to DCPO and DPO. These outcomes align with the objectives of each method, where MaPO's margin-oriented formulation is sensitive to over-weighting the margin and thus favors small $\beta$, DCPO's diffusion-consistent objective with dual captions scales the preference signal while preserving stable updates and thus benefits from larger $\beta$, and Diffusion-DPO's balance between the preference term and a reference-model KL naturally favors a moderate $\beta$. For additional details on the hyperparameter tuning process refer to Appendix F.

**DCPO-h vs Diffusion-DPO on GPT-4o Judgment.** We evaluated DCPO-h and Diffusion-DPO using GPT-4o on the PartiPrompts benchmark, consisting of 1,632 prompts. GPT-4o assessed images based on three criteria: **Q1)** General Preference (*Which image do you prefer given the prompt?*), **Q2)** Visual Appeal (*Which image is more visually appealing?*), and **Q3)** Prompt Alignment (*Which image better fits the text description?*). As shown in Figure 7, DCPO-h outperformed Diffusion-DPO in Q1 and Q2, with win rates of 58% and 66%. Beyond GPT-4o, we conducted a small human study for Q2 to measure the alignment of GPT-4o with human preference; results in Table 14 in Appendix G indicate that GPT-4o's judgments align closely with human preferences. See Appendix G for more details.

**Alternative MLLM-as-a-Judge for Controllable Perturbation.** We use CLIPScore as the selection mechanism for incorporating the three levels of perturbations used in our PickDouble dataset. To confirm the effectiveness of CLIPScore-as-a-Judge on perturbation selection, we further conducted a series of controlled

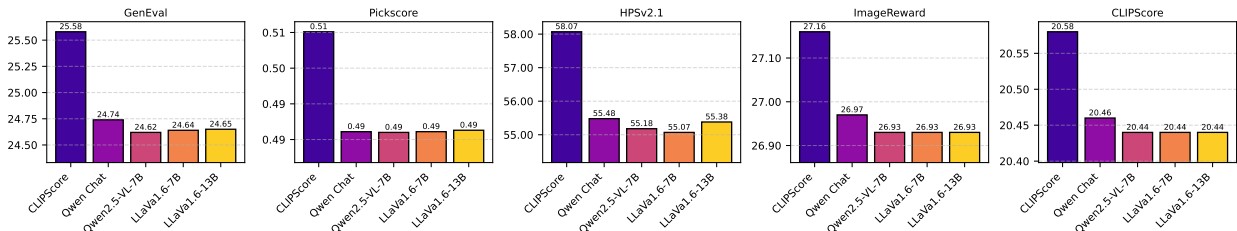

Figure 7: **(Left)** PartiPrompts benchmark results for three evaluation questions, as voted by GPT-4o. **(Right)** Qualitative comparison between DCPO-h and Diffusion-DPO fine-tuned on SD 2.1. DCPO-h shows better prompt adherence and realism, with outputs that align more closely with human preferences, emphasizing high contrast, vivid colors, fine detail, and well-focused composition.

Figure 8: Benchmarking results by alternative perturbation selection mechanisms (MLLM-as-a-Judge models) alongside our original CLIPScore-based judge. All experiments are under the same SD 2.1 and DCPO-h + LLaVa captions setting. For clarity, we only show the averages across all perturbation levels, per perturbation judge and benchmark.

ablation studies using perturbations ranked by alternative MLLM-based judges inspired by Chen et al. (2024). We experimented with the following judge models - Qwen Chat, Qwen2.5-VL-7B, LlaVa1.6-7B, and LlaVa1.6-13B. The results in Figure 8 show that all MLLM-as-a-Judge variants consistently underperform compared to our CLIPScore-based baselines across all five downstream benchmarks, regardless of the judge model's scale or the granularity of its perturbation strengths. This further validates that our original method identifies more effective perturbations (i.e., modified prompts) that lead to improved downstream performance across multiple benchmarks. More details can be found in Appendix E.

**DCPO vs Online and Iterative Methods.** Although DCPO is an offline method, we compare it against online and iterative approaches such as SePPO Zhang et al. (2024) and SPO Liang et al. (2025) to assess effectiveness. Under a small budget of optimization steps and limited training data, DCPO consistently outperforms both, demonstrating superior sample and compute efficiency. By contrast, SPO typically requires substantially more training iterations and data to reach similar performance, and SePPO depends on iterative online updates to approach comparable results. These findings indicate that DCPO achieves strong alignment with fewer resources while avoiding the complexity of online optimization. Refer to Appendix B.3 for more details.

# 5 Related Works

**Aligning Diffusion Models.** Recent advances in preference alignment for text-to-image diffusion models show that RL-free methods (Yang et al., 2024a; Li et al., 2024; Yuan et al., 2024; Gambashidze et al., 2024; Park et al., 2024) outperform RL-based ones (Fan & Lee, 2023; Fan et al., 2023; Hao et al., 2023; Lee et al., 2023; Prabhudesai et al., 2024; Black et al., 2024; Clark et al., 2024) by removing the need for explicit reward models. Methods like Diffusion-DPO (Wallace et al., 2024), which adapts DPO (Rafailov et al., 2024), and Diffusion-KTO (Li et al., 2024), which uses binary feedback instead of pairwise data, streamline alignment. MaPO (Hong et al., 2024) further increases flexibility by eliminating the dependency on reference models.

However, these methods often align based on single-prompt image pairs, leading to issues with irrelevant prompts (see Section 3.1). While online preference optimization shows strong performance (Fan et al., 2023; Yang et al., 2024b; Lou et al., 2024), it requires a reward model and more optimization steps, making it vulnerable to reward hacking. In this work, we focus on comparing DCPO with offline methods that do not rely on extra models for ranking of diffusion outputs.

**Text-to-image Preference Datasets.** Text-to-image image preference datasets commonly involve the text prompt to generate the images, and two or more images are ranked according to human preference. HPS (Wu et al., 2023c) and HPSv2 (Wu et al., 2023b) create multiple images using a series of image generation models for a single prompt, and the images are ranked according to real-world human preferences. Moreover, a classifier is trained using the gathered preference dataset, which can be used as a metric for image-aligning tasks. Also, Pick-a-Pic v2 (Kirstain et al., 2023) follows a similar structure to create a pairwise preference dataset along with their CLIP (Radford et al., 2021) based scoring function, Pickscore. While these datasets are carefully created, having only one prompt for both or all the images introduces *semantic overlap*, which will be further discussed in Section 3.1. For this reason, we modified the Pick-a-Pic v2 dataset using recaptioning and perturbation methods to improve image alignment performance.

## 6 Conclusion

In this paper, we present a novel preference optimization method for aligning text-to-image diffusion models called Dual Caption Preference Optimization (DCPO). We tackle two major challenges in previous preference datasets and optimization algorithms: the *semantic overlap* and *irrelevant prompt*. To overcome these issues, we introduce the *Pick-Double Caption* dataset, a modified version of the Pick-a-Pic v2 dataset. We also identify difficulties in generating captions, particularly the risk of out-of-distribution captions for images, and propose three approaches: 1) captioning (DCPO-c), 2) perturbation (DCPO-p), and 3) a hybrid method (DCPO-h). Our results show that DCPO-h significantly enhances alignment performance, outperforming methods like MaPO and Diffusion-DPO across multiple metrics.

### Limitation

While DCPO achieves strong performance on various benchmarking metrics, its captioning and perturbation processes require considerable computational resources. We encourage future research to explore more efficient and cost-effective alternatives to further enhance its practicality. The preliminary results in Appendix B.5 indicate that DCPO outperforms Diffusion-DPO across different backbones, such as Stable Diffusion XL (SDXL) (Podell et al., 2023). However, further investigation into its performance with emerging state-of-the-art models remains essential. Additionally, exploring DCPO's applications in safety-related tasks could be a valuable direction for future work. We believe our research will contribute significantly to the alignment community and inspire further advancements in this field.

### Ethical Considerations

The authors state that in this work, AI assistants, specifically Grammarly and ChatGPT, were utilized to correct grammatical errors and restructure sentences.

### Acknowledgments

We thank the Research Computing (RC) at Arizona State University (ASU) and cr8dl.ai for their generous support in providing computing resources. The views and opinions of the authors expressed herein do not necessarily state or reflect those of the funding agencies and employers.

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

# A Formalized Proofs of DCPO

## A.1 Optimizing the DCPO Loss is Optimizing the DPO Loss

Inspired by Bansal et al. (2024), we can intuitively assume that the DCPO objective is to learn an aligned model $p_\theta$ by weighting the joint probability of preferred images $p_\theta(x_0^w, z^w)$ over less preferred images $p_\theta(x_0^l, z^l)$. We set the optimization objective of DCPO is to minimize the following:

$$\mathcal{L}_{\text{DCPO}}(\theta) = -\mathbb{E}_{(x_0^w, x_0^l, z^l, z^w) \sim \mathcal{D}'} \log \sigma(\beta \mathbb{E}_{x_{1:T}^w \sim p_\theta(x_{1:T}^w | x_0^w, z^w), x_{1:T}^l \sim p_\theta(x_{1:T}^l, z^l)} \\ [\log \frac{p_\theta(x_{0:T}^w, z^w)}{p_{\text{ref}}(x_{0:T}^w, z^w)} - \log \frac{p_\theta(x_{0:T}^l, z^l)}{p_{\text{ref}}(x_{0:T}^l | z^l)}]) \quad (8)$$

Here, we highlight that reducing $\mathcal{L}_{\text{DCPO}}(\theta)$ is equivalently reducing $\mathcal{L}_{\text{DPO}}(\theta)$ when the captions are the same for the preferred and less preferred images.

**Lemma 1.** *Under the case where $\mathcal{D}_{define} = \{x_0^w, c, x_0^l, c\}$, that is, the image captions are identical for the given pair of preferred and less preferred images $(x_0^w, x_0^l)$, we have $L_{DPO}(\theta; \mathcal{D}_{DPO}; \beta; p_{ref}) = L_{DCPO}(\theta; \mathcal{D}_{define}; \beta; p_{ref})$, in which $\mathcal{D}_{DPO} = \{c, x_0^w, x_0^l\}$.*

**Proof of Lemma 1.**

$$\mathcal{L}_{\text{DCPO}}(\theta; \mathcal{D}', \beta, p_{\text{ref}}) = \mathbb{E}_{(x_0^w, x_0^l, z^w, z^l) \sim \mathcal{D}'} \\ \left[ \log \left( \sigma \left( \beta \log \frac{p_\theta(x_0^w, z^w)}{p_{\text{ref}}(x_0^w, z^w)} - \beta \log \frac{p_\theta(x_0^l, z^l)}{p_{\text{ref}}(x_0^l, z^l)} \right) \right) \right] \\ = \mathbb{E}_{(x_0^w, x_0^l, z^w, z^l) \sim \mathcal{D}'} \\ \left[ \log \left( \sigma \left( \beta \log \frac{p_\theta(x_0^w | z^w) p_\theta(z^w)}{p_{\text{ref}}(x_0^w | z^w) p_{\text{ref}}(z^w)} - \beta \log \frac{p_\theta(x_0^l | z^l) p_\theta(z^l)}{p_{\text{ref}}(x_0^l | z^l) p_{\text{ref}}(z^l)} \right) \right) \right] \quad (9)$$

$$\mathcal{L}_{\text{DCPO}}(\theta; \mathcal{D}_{\text{define}}, \beta, p_{\text{ref}}) \overset{z^w = z^l = c}{=} \mathbb{E}_{(x_0^w, c, x_0^l, c) \sim \mathcal{D}_{\text{define}}} \\ \left[ \log \left( \sigma \left( \frac{p_\theta(c)}{p_{\text{ref}}(c)} \left( \beta \log \frac{p_\theta(x_0^w | c)}{p_{\text{ref}}(x_0^w | c)} - \beta \log \frac{p_\theta(x_0^l | c)}{p_{\text{ref}}(x_0^l | c)} \right) \right) \right) \right] \\ \overset{\frac{p_\theta(c)}{p_{\text{ref}}(c)} = C}{=} \mathbb{E}_{(x_0^w, x_0^l, c) \sim \mathcal{D}_{\text{DPO}}} \\ \left[ \log \left( \sigma \left( C \cdot \beta \log \frac{p_\theta(x_0^w | c)}{p_{\text{ref}}(x_0^w | c)} - C \cdot \beta \log \frac{p_\theta(x_0^l | c)}{p_{\text{ref}}(x_0^l | c)} \right) \right) \right] \\ = \mathcal{L}_{\text{DPO}}(\theta; \mathcal{D}_{\text{DPO}}, \beta, p_{\text{ref}}) \quad (10)$$

In Equation 10, $C$ is a constant value that equates to $\frac{p_\theta(c)}{p_{\text{ref}}(c)}$. The proof above follows the Bayes rule by substituting $c$ according to $z^w = z^l = c$.

## A.2 Analyses of DCPO's Effectiveness

In this section, we present the formal proofs of why our DCPO leads to a more optimized $L(\theta)$ of a Diffusion-based model and, consequently, better performance in preference alignment tasks.

**Proof 1.** *Increasing the difference between $\Delta_{preferred}$ and $\Delta_{less\text{-}preferred}$ improves the optimization of $L(\theta)$.* For better clarity, the loss function $L(\theta)$ can be written as:

$$L(\theta) = -\mathbb{E} \left[ \log \sigma \left( -\beta T \omega(\lambda_t) \cdot M \right) \right]$$

where $\sigma(x)$ is the sigmoid function that squashes its input $x$ into the output range $(0,1)$, and $M = \Delta_{\text{preferred}} - \Delta_{\text{less-preferred}}$, i.e., the margin between the respective importance of the preferred and less preferred predictions.

Characteristically, the gradient of $\sigma(x)$ is at its maximum near $x = 0$ and decreases as $|x|$ increases. A larger margin in terms of $M$ makes it easier for the optimization to drive the sigmoid function towards its asymptotes, reducing loss.

- When $M$ is small ($|M| \approx 0$): The sigmoid $\sigma(-\beta T \omega(\lambda_t) \cdot M)$ is near 0.5 (its midpoint). Also, the gradient of $\log \sigma(x)$ is the largest near this point, meaning the model struggles to differentiate between preferred and less preferred predictions effectively.

- When $M$ is large ($|M| \gg 0$): The sigmoid $\sigma(-\beta T \omega(\lambda_t) \cdot M)$ moves closer to 0 or 1, depending on the sign of $M$. For a well-aligned model, if the preferred predictions are correct, $M > 0$ and $\sigma(-\beta T \omega(\lambda_t) \cdot M)$ approach 1, thus minimizing the loss.

Intuitively, an ideally large $M$ represents a clear distinction between the preferred image-caption versus the less preferred image-caption. Thus, by maximizing $M$, we may push the loss $L(\theta)$ towards its minimum, leading to better soft-margin optimization.

**Proof 2.** *Replacing caption c with the specifically generated caption $z^l$ for the less-preferred image $\mathbf{x}_0^l$ decreases $\Delta_{less\text{-}preferred}$.*

To analyze how replacing $\mathbf{c}$ with $\mathbf{z}^l$, where $\mathbf{c} \subset \mathbf{z}^l$ and $\mathbf{z}^l \sim Q(\mathbf{z}^l|x^l, c)$, for the less-preferred image $\mathbf{x}_0^l$ improves the optimization, we delve into how the loss function is affected by this substitution.
The term relevant to the less-preferred image $\mathbf{x}_t^l$ in the loss is:

$$\Delta_{\text{less-preferred}} = \|\boldsymbol{\epsilon}^l - \boldsymbol{\epsilon}_\theta(\mathbf{x}_t^l, t, \mathbf{c})\|_2^2 - \|\boldsymbol{\epsilon}^l - \boldsymbol{\epsilon}_{\text{ref}}(\mathbf{x}_t^l, t, \mathbf{c})\|_2^2.$$

Replacing $\mathbf{c}$ with $\mathbf{z}^l$ modifies the predicted noise term $\boldsymbol{\epsilon}_\theta(\mathbf{x}_t^l, t, \mathbf{c})$ to $\boldsymbol{\epsilon}_\theta(\mathbf{x}_t^l, t, \mathbf{z}^l)$. Since $\mathbf{z}^l$ better represents $\mathbf{x}_t^l$, we have:

$$\|\boldsymbol{\epsilon}^l - \boldsymbol{\epsilon}_\theta(\mathbf{x}_t^l, t, \mathbf{z}^l)\|_2^2 < \|\boldsymbol{\epsilon}^l - \boldsymbol{\epsilon}_\theta(\mathbf{x}_t^l, t, \mathbf{c})\|_2^2 \tag{11}$$

When $\|\boldsymbol{\epsilon}^l - \boldsymbol{\epsilon}_\theta(\mathbf{x}_t^l, t, \mathbf{z}^l)\|_2^2$ becomes smaller, the term $\Delta_{\text{less-preferred}}$ decreases. This leads to $\Delta_{\text{preferred}} - \Delta_{\text{less-preferred}}$ becoming larger, which improves the soft-margin optimization in the loss function $L(\theta)$ that we have shown in Proof 1.

We further elaborate on why Equation 11 is true. In the context of mean squared error (MSE) minimization, the optimal predictor of $\boldsymbol{\epsilon}^l$ given some information is the conditional expectation:

- When conditioned on $(\mathbf{x}_t^l, t, c)$:
$$\boldsymbol{\epsilon}_\theta^*(\mathbf{x}_t^l, t, c) = \mathbb{E}\left[\boldsymbol{\epsilon}^l \mid \mathbf{x}_t^l, t, c\right]$$

- When conditioned on $(\mathbf{x}_t^l, t, z^l)$:
$$\boldsymbol{\epsilon}_\theta^*(\mathbf{x}_t^l, t, z^l) = \mathbb{E}\left[\boldsymbol{\epsilon}^l \mid \mathbf{x}_t^l, t, z^l\right]$$

The total variance of $\boldsymbol{\epsilon}^l$ can be decomposed as by the Law of Total Variance (conditional variance formula) (Ross, 2014):

$$\text{Var}\left(\boldsymbol{\epsilon}^l\right) = \mathbb{E}\left[\text{Var}\left(\boldsymbol{\epsilon}^l \mid \mathbf{x}_t^l, t, c\right)\right] + \text{Var}\left(\mathbb{E}\left[\boldsymbol{\epsilon}^l \mid \mathbf{x}_t^l, t, c\right]\right)$$

Similarly, when conditioning on $z^l$:

$$\text{Var}\left(\boldsymbol{\epsilon}^l\right) = \mathbb{E}\left[\text{Var}\left(\boldsymbol{\epsilon}^l \mid \mathbf{x}_t^l, t, z^l\right)\right] + \text{Var}\left(\mathbb{E}\left[\boldsymbol{\epsilon}^l \mid \mathbf{x}_t^l, t, z^l\right]\right)$$

Since $c \subset z^l$, the information provided by $z^l$ is richer than that of $c$. In probability theory, conditioning on more information does not increase the conditional variance:

$$\text{Var}\left(\boldsymbol{\epsilon}^l \mid \mathbf{x}_t^l, t, z^l\right) \leq \text{Var}\left(\boldsymbol{\epsilon}^l \mid \mathbf{x}_t^l, t, c\right) \tag{12}$$

This inequality holds because conditioning on additional information $(z^l)$ can only reduce or leave unchanged the uncertainty (variance) about $\boldsymbol{\epsilon}^l$.

The expected squared error when using the optimal predictor is equal to the conditional variance:

$$\mathbb{E}\left[\left\|\boldsymbol{\epsilon}^l - \boldsymbol{\epsilon}_\theta^*(\mathbf{x}_t^l, t, c)\right\|_2^2\right] = \mathbb{E}\left[\text{Var}\left(\boldsymbol{\epsilon}^l \mid \mathbf{x}_t^l, t, c\right)\right]$$

Similarly,

$$\mathbb{E}\left[\left\|\boldsymbol{\epsilon}^l - \boldsymbol{\epsilon}_\theta^*(\mathbf{x}_t^l, t, z^l)\right\|_2^2\right] = \mathbb{E}\left[\text{Var}\left(\boldsymbol{\epsilon}^l \mid \mathbf{x}_t^l, t, z^l\right)\right]$$

From 12, we have:

$$\text{Var}\left(\boldsymbol{\epsilon}^l \mid \mathbf{x}_t^l, t, z^l\right) \leq \text{Var}\left(\boldsymbol{\epsilon}^l \mid \mathbf{x}_t^l, t, c\right)$$

Taking expectations on both sides:

$$\mathbb{E}\left[\text{Var}\left(\boldsymbol{\epsilon}^l \mid \mathbf{x}_t^l, t, z^l\right)\right] \leq \mathbb{E}\left[\text{Var}\left(\boldsymbol{\epsilon}^l \mid \mathbf{x}_t^l, t, c\right)\right]$$

Therefore,

$$\mathbb{E}\left[\left\|\boldsymbol{\epsilon}^l - \boldsymbol{\epsilon}_\theta^*(\mathbf{x}_t^l, t, z^l)\right\|_2^2\right] \leq \mathbb{E}\left[\left\|\boldsymbol{\epsilon}^l - \boldsymbol{\epsilon}_\theta^*(\mathbf{x}_t^l, t, c)\right\|_2^2\right]$$

Assuming that the neural network $\boldsymbol{\epsilon}_\theta$ is capable of approximating the optimal predictor $\boldsymbol{\epsilon}_\theta^*$, especially as training progresses and the model capacity is sufficient, we can write:

$$\left\|\boldsymbol{\epsilon}^l - \boldsymbol{\epsilon}_\theta(\mathbf{x}_t^l, t, z^l)\right\|_2^2 \approx \left\|\boldsymbol{\epsilon}^l - \boldsymbol{\epsilon}_\theta^*(\mathbf{x}_t^l, t, z^l)\right\|_2^2$$

Similarly for $c$

$$\left\|\boldsymbol{\epsilon}^l - \boldsymbol{\epsilon}_\theta(\mathbf{x}_t^l, t, c)\right\|_2^2 \approx \left\|\boldsymbol{\epsilon}^l - \boldsymbol{\epsilon}_\theta^*(\mathbf{x}_t^l, t, c)\right\|_2^2 .$$

Therefore, the expected squared error satisfies:

$$\mathbb{E}\left[\left\|\boldsymbol{\epsilon}^l - \boldsymbol{\epsilon}_\theta(\mathbf{x}_t^l, t, z^l)\right\|_2^2\right] \leq \mathbb{E}\left[\left\|\boldsymbol{\epsilon}^l - \boldsymbol{\epsilon}_\theta(\mathbf{x}_t^l, t, c)\right\|_2^2\right]$$

Since the term of $\Delta_{\text{less-preferred}}$ in the loss function involves the difference of squared errors, using $z^l$ instead of $c$ for the less preferred sample results in a lower error term:

$$\Delta_{\text{less-preferred}}^{(z^l)} = \left\|\boldsymbol{\epsilon}^l - \boldsymbol{\epsilon}_\theta(\mathbf{x}_t^l, t, z^l)\right\|_2^2 - \left\|\boldsymbol{\epsilon}^l - \boldsymbol{\epsilon}_{\text{ref}}(\mathbf{x}_t^l, t, z^l)\right\|_2^2$$

Comparing with the original:

$$\Delta_{\text{less-preferred}}^{(c)} = \left\|\boldsymbol{\epsilon}^l - \boldsymbol{\epsilon}_\theta(\mathbf{x}_t^l, t, c)\right\|_2^2 - \left\|\boldsymbol{\epsilon}^l - \boldsymbol{\epsilon}_{\text{ref}}(\mathbf{x}_t^l, t, c)\right\|_2^2$$

Assuming the reference model $\boldsymbol{\epsilon}_{\text{ref}}$ remains the same or also benefits similarly from the additional information in $z^l$, the net effect is that the first term decreases more than the second term, leading to a reduced $\Delta_{\text{less-preferred}}$.

**Proof 3** *Replacing caption $c$ with the specifically generated caption $z^w$ for the preferred image $\mathbf{x}_0^w$ increases $\Delta_{preferred}$.*

To prove that replacing $\mathbf{c}$ with $\mathbf{z}^w \sim Q(z^w|x^w, c)$, where $\mathbf{c} \subset \mathbf{z}^w$, for $\mathbf{x}_0^w$ also contributes to a better optimized loss $L(\theta)$, we examine how this particular substitution affects the loss function.
We let

$$R_\theta(\mathbf{c}) = \|\boldsymbol{\epsilon}^w - \boldsymbol{\epsilon}_\theta(\mathbf{x}_t^w, t, \mathbf{c})\|_2^2,$$

$$R_{\text{ref}}(\mathbf{c}) = \|\boldsymbol{\epsilon}^w - \boldsymbol{\epsilon}_{\text{ref}}(\mathbf{x}_t^w, t, \mathbf{c})\|_2^2.$$

The rate of decrease in $R_\theta$ due to $\mathbf{z}^w$ is proportional to the model's ability to exploit the additional conditioning. Since $\boldsymbol{\epsilon}_\theta$ is learnable, it can more effectively leverage $\mathbf{z}^w$ than $\boldsymbol{\epsilon}_{\text{ref}}$, yielding:

$$\Delta R_\theta = R_\theta(\mathbf{c}) - R_\theta(\mathbf{z}^w) \gg \Delta R_{\text{ref}} = R_{\text{ref}}(\mathbf{c}) - R_{\text{ref}}(\mathbf{z}^w).$$

We further elaborate on why the learnable model's noise prediction residual ($R_\theta$) decreases faster than the reference model's residual ($R_{\text{ref}}$) when $\mathbf{c}$ is replaced by $\mathbf{z}^w$. The residuals for the learnable and reference models are defined as:

$$R_\theta(\mathbf{c}) = \|\boldsymbol{\epsilon}^w - \boldsymbol{\epsilon}_\theta(\mathbf{x}_t^w, t, \mathbf{c})\|_2^2,$$

$$R_{\text{ref}}(\mathbf{c}) = \|\boldsymbol{\epsilon}^w - \boldsymbol{\epsilon}_{\text{ref}}(\mathbf{x}_t^w, t, \mathbf{c})\|_2^2.$$

When $\mathbf{c}$ is replaced with $\mathbf{z}^w$ (where $\mathbf{c} \subset \mathbf{z}^w$), the residuals become:

$$R_\theta(\mathbf{z}^w) = \|\boldsymbol{\epsilon}^w - \boldsymbol{\epsilon}_\theta(\mathbf{x}_t^w, t, \mathbf{z}^w)\|_2^2,$$

$$R_{\text{ref}}(\mathbf{z}^w) = \|\boldsymbol{\epsilon}^w - \boldsymbol{\epsilon}_{\text{ref}}(\mathbf{x}_t^w, t, \mathbf{z}^w)\|_2^2.$$

The rate of decrease for each residual is defined as:

$$\Delta R_\theta = R_\theta(\mathbf{c}) - R_\theta(\mathbf{z}^w),$$

$$\Delta R_{\text{ref}} = R_{\text{ref}}(\mathbf{c}) - R_{\text{ref}}(\mathbf{z}^w).$$

The quality of conditioning, $Q(\mathbf{c})$, represents how well the conditioning $\mathbf{c}$ aligns with the true noise $\boldsymbol{\epsilon}^w$. We assume that

$$Q(\mathbf{z}^w) > Q(\mathbf{c}),$$

where the improvement in conditioning quality $\Delta Q$ is defined as

$$\Delta Q = Q(\mathbf{z}^w) - Q(\mathbf{c}).$$

The residual for $R_\theta$ is proportional to the misalignment between $Q(\mathbf{c})$ and $\boldsymbol{\epsilon}^w$:

$$R_\theta(\mathbf{c}) \propto \frac{1}{Q(\mathbf{c})}.$$

Replacing $\mathbf{c}$ with $\mathbf{z}^w$ (higher $Q$) results in a larger proportional reduction:

$$R_\theta(\mathbf{z}^w) \propto \frac{1}{Q(\mathbf{z}^w)} \quad \text{with} \quad \Delta R_\theta \propto \Delta Q.$$

The reference model's residual $R_{\mathrm{ref}}$ depends weakly on $Q(\mathbf{c})$, as it is fixed or less adaptable:

$$R_{\mathrm{ref}}(\mathbf{c}) \propto \frac{1}{Q_{\mathrm{ref}}(\mathbf{c})},$$

where $Q_{\mathrm{ref}}(\mathbf{c})$ is less sensitive to changes in $\mathbf{c}$.

Thus, the proportional improvement in $R_\theta$ due to $\Delta Q$ is significantly larger than for $R_{\mathrm{ref}}$.

The preferred difference term is:

$$\Delta_{\mathrm{preferred}} = R_\theta - R_{\mathrm{ref}}.$$

As $R_\theta$ decreases significantly more than $R_{\mathrm{ref}}$, the gap $R_\theta - R_{\mathrm{ref}}$ becomes larger, increasing $\Delta_{\mathrm{preferred}}$:

$$\Delta R_\theta \gg \Delta R_{\mathrm{ref}} \implies \Delta_{\mathrm{preferred}} \text{ increases.}$$

The learnable model $\boldsymbol{\epsilon}_\theta$ benefits more from the improved conditioning $\mathbf{z}^w$ because of its adaptability and training dynamics. This results in a larger reduction in $R_\theta$ compared to $R_{\mathrm{ref}}$. Mathematically, the relative rate of decrease:

$$\text{Relative Rate} = \frac{\Delta R_\theta}{\Delta R_{\mathrm{ref}}} \gg 1,$$

which ensures that $\Delta_{\mathrm{preferred}}$ also increases, hence improving the optimization process in $L(\theta)$ and helping the model distinguish predictions on preferred and less preferred image-captions more effectively.

## B Supplementary Experiments and Analyses on DCPO

### B.1 *Semantic Overlap Distribution* Measured in VQAScore

We also investigated the *conflict distribution* challenge of preference optimization datasets described in Section 3.1 using more recent prompt-image alignment measures, such as VQAScore (Karthik et al., 2023; Huang et al., 2023). Likewise, we calculated the VQAScores between each prompt $c$ and its preferred image, as well as between each prompt and its less-preferred image, from the Pick-a-Pic V2 dataset. Our results in Figure 9 indicate that, similar to the counterpart in terms of CLIPScore in Figure 3, there exists a consistently significant conflict between the semantic distributions of the preferred and less-preferred images with respect to the prompt $c$.

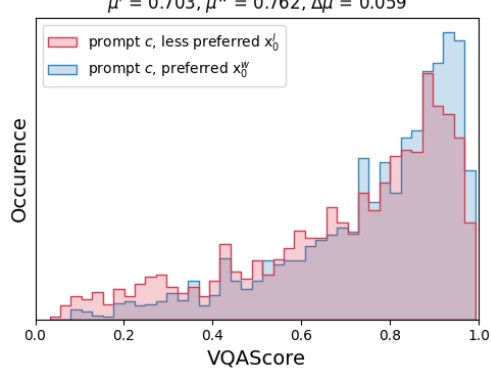

Figure 9: The *conflict distribution* that exists in the Pick-a-Pic V2 dataset in terms of VQAScore.

### B.2 Comparison with Diffusion-KTO

A preference alignment dataset, such as Pick-a-Pic (Kirstain et al., 2023), is defined as $D = \{c, x^w, x^l\}$, where $x^w$ and $x^l$ represent the preferred and less preferred images for the prompt $c$. Diffusion-KTO (Li et al., 2024) hypothesizes the optimization of a diffusion model using

Table 5: Comparison of DCPO-h and Diffusion-KTO across various benchmarks.

| Method | GenEval (↑) | Pickscore (↑) | HPSv2.1 (↑) | ImageReward (↑) | CLIPscore (↑) |
|---|---|---|---|---|---|
| Diffusion-KTO | 0.5008 | 20.41 | 24.80 | 55.5 | 26.95 |
| DCPO-h | **0.5100** | **20.57** | **25.62** | **58.2** | **27.13** |

only a single preference label based on whether an image $x$ is suitable or not for a given prompt $c$. Diffusion-KTO uses a differently formatted input dataset $D = \{c, x\}$, where $x$ is a generated image corresponding to the prompt $c$.

Diffusion-KTO's hypothesis is fundamentally different from our DCPO's. While Diffusion-KTO focuses on binary preferences (like/dislike) for individual image-prompt pairs, our approach involves paired preferences. We observe that using the same prompt $c$ for both preferred and less preferred images may not be ideal. To address this, we propose optimizing a diffusion model using a dataset in terms of $D = \{z^w, z^l, x^w, x^l\}$, where $z^w$ and $z^l$ are the captions generated by a static captioning model $Q_\phi$ for the preferred and less preferred images, respectively, referring to the original prompt.

We nonetheless conduct comparisons between Diffusion-KTO and DCPO on various preference alignment benchmarks. The results in Table 5 show that our DCPO-h consistently outperforms Diffusion-KTO on all benchmarks, demonstrating the effectiveness of our DCPO method.

## B.3 Comparison with Online and Interactive Methods

Beyond offline preference optimization methods such as Diffusion-DPO, MaPO, and Diffusion-KTO, we also evaluate iterative algorithms (e.g., SePPO) and online algorithms (e.g., SPO). Under a fixed training budget of 2,000 steps and a dataset of 20,000 samples, the results in Table 6 show that DCPO outperforms both SPO and SePPO, indicating greater cost efficiency and stronger alignment with fewer optimization steps and limited data. We leave a comprehensive comparison of offline and online methods under larger training budgets and datasets for future work.

Table 6: Comparison of DCPO-h against online (SPO) and iterative (SePPO) methods under a fixed training budget.

| Method | GenEval (↑) | Pickscore (↑) | HPSv2.1 (↑) | ImageReward (↑) | CLIPscore (↑) |
|---|---|---|---|---|---|
| SPO | 0.4449 | 19.30 | 24.40 | 46.45 | 25.91 |
| SePPO (lr $1 \times 10^{-8}$) | 0.50567 | 20.34 | 25.09 | 55.21 | 26.83 |
| SePPO (lr $5 \times 10^{-9}$) | 0.50232 | 20.32 | 25.08 | 55.24 | 26.82 |
| DCPO-h | **0.5100** | **20.57** | **25.62** | **58.20** | **27.13** |

## B.4 Benchmarking Performance on Rapidata

To further demonstrate DCPO's versatility, we fine-tune Stable Diffusion 2.1 using Diffusion-DPO and DCPO on another high-quality preference dataset Rapidata (Christodoulou & Kuhlmann-Jørgensen, 2024). Table 7 shows that our DCPO variants consistently outperform the Diffusion-DPO baseline on multiple benchmarking metrics, including GenEval, Pickscore, HPSv2.1, ImageReward, and CLIPscore.

Table 7: DCPO performances using SD2.1 on Rapidata (Christodoulou & Kuhlmann-Jørgensen, 2024).

| Method (SD2.1) | Geneval | Pickscore | HPSv2.1 | ImageReward | CLIPscore |
|---|---|---|---|---|---|
| Diffusion-DPO | 0.4813 | 20.34 | 25.10 | 55.4 | 26.84 |
| DCPO-c | 0.4867 | **20.44** | **25.43** | **55.7** | 26.86 |
| DCPO-h | **0.4978** | 20.42 | 25.10 | 55.6 | **26.91** |

### B.5 Performance of Using SDXL as the Backbone of DCPO

We also perform additional experiments to evaluate the performance of DCPO using Stable Diffusion XL (SDXL) (Podell et al., 2023), a larger alternative backbone model for T2I instead of our default SD 2.1. Due to the larger parameter size of SDXL, we conduct LoRA fine-tuning with minimal hyperparameter search. The results in Table 8 show that DCPO outperforms Diffusion-DPO on key metrics such as Pickscore, Geneval, HPSv2, and CLIPscore, while achieving comparable performance on ImageReward. We hope that our results would encourage other researchers to further explore DCPO's effectiveness on different datasets in future, gaining more valuable insights into its broader applicability and robustness.

Table 8: DCPO performances of using Stable Diffusion XL (SDXL) (Podell et al., 2023) as the backbone model.

| Method (SDXL) | Geneval (Overall) | Pickscore | HPSv2.1 | ImageReward | CLIPscore |
|---|---|---|---|---|---|
| Diffusion-DPO | 0.5645 | 21.77 | 28.64 | 71.2 | 28.61 |
| DCPO-c | **0.5758** | **21.87** | **28.65** | 71.2 | **28.63** |
| DCPO-h-weak | 0.5704 | **21.87** | 28.64 | 71.2 | 28.62 |
| DCPO-h-medium | 0.5700 | 21.86 | 28.64 | 71.2 | **28.63** |
| DCPO-h-strong | 0.5696 | 21.86 | 28.64 | 71.2 | 28.62 |

## C  Pick-Double Caption Dataset

In this section, we provide details about the *Pick-Double Caption* dataset. As discussed in Section 4.1, we sampled 20,000 instances from the Pick-a-Pic v2 dataset and excluded those with equal preference scores. We plot the distribution of the original prompts, as shown in Figure 10.

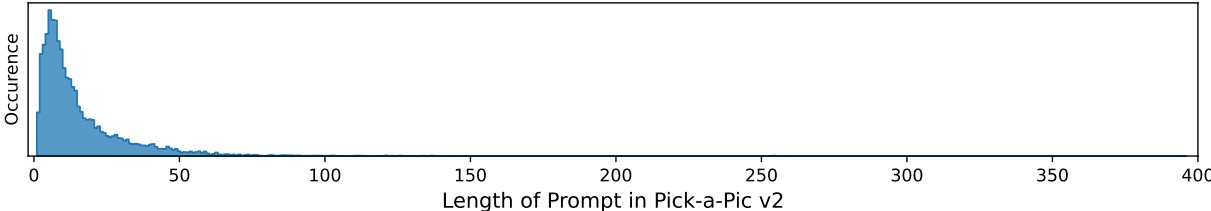

Figure 10: The distribution of token lengths in the original prompts.

We observed that some prompts contained only one or two words, while others were excessively long. To ensure a fair comparison, we removed prompts that were too short or too long, leaving us with approximately 17,000 instances. We then generated captions using two state-of-the-art models, LLaVA-1.6-34B, and Emu2-32B. The construction of the *Pick-Double Caption* dataset is illustrated in Figure 11, which provides several examples.

We acknowledge that while captioning and perturbation introduce additional computational costs, these are one-time expenses incurred only during pre-processing and do not affect training or evaluation time. We introduce three variants of DCPO—c, p, and h—each with different processing requirements. For instance, DCPO-c and DCPO-h involve captioning, whereas DCPO-p is more computationally efficient as it only perturbs a less preferred caption. Furthermore, captioning 20,000 images using LLaVA requires less than 12 hours on a single A100:80G GPU, and this process can be significantly accelerated using multiple GPUs.

As explained in Section 4.1, we utilized two types of prompts to generate captions: 1) Conditional prompt and 2) Non-conditional prompt. Below, we outline the specific prompts used for each captioning method.

**Example of Conditional Prompt**

Using one sentence, describe the image based on the following prompt: *playing chess tournament on the moon.*

**Example of Non-Conditional Prompt**

Using one sentence, describe the image.

Table 9 presents a statistical analysis of the *Pick-Double Caption* dataset. With the non-conditional prompt method, we found that the average token length of captions generated by LLaVA is similar to that of the original prompts. However, captions generated by LLaVA using conditional prompts are twice as long as the original prompts. Additionally, Emu2 generated captions that, on average, are half the length of the original prompts for both methods.

Table 9: Statistical information on the Pick-Double Caption dataset, including the CLIPscore of in-distribution data and average token count of captions generated by LLaVA and Emu2 for both in-distribution and out-of-distribution data.

| Text | Token Len. (Avg-in) | Token Len. (Avg-out) | CLIP score (in) | CLIP score (out) |
|---|---|---|---|---|
| prompt $c$ | 15.95 | 15.95 | (26.74, 25.41) | (26.74, 25.41) |
| caption $z^w$ (LLaVA) | 32.32 | 17.69 | 30.85 | 29.04 |
| caption $z^l$ (LLaVA) | 32.83 | 17.91 | 26.48 | 28.29 |
| caption $z^w$ (Emu2) | 7.75 | 8.40 | 25.44 | 25.18 |
| caption $z^l$ (Emu2) | 7.84 | 8.44 | 22.64 | 24.88 |

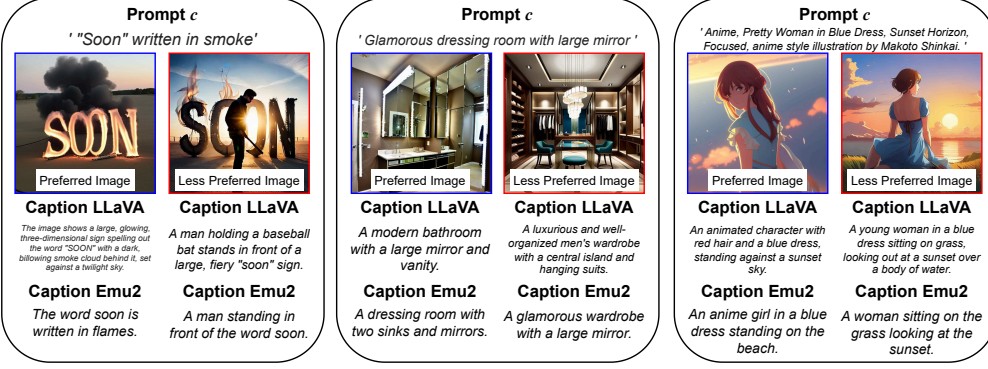

Figure 11: Examples of Pick-Double Caption dataset.

# D    Specifications of Creating Perturbed Captions

We provide the setups for the LLM-based perturbation process involved in the DCPO-p and DCPO-h pipelines. Similarly to the method of constructing paraphrasing adversarial attacks as synonym-swapping perturbation by Krishna et al. (2024), we use DIPPER, a text generation model built by fine-tuning T5-XXL (Chung et al., 2022), to create semantically perturbed captions or prompts, as shown in Table 10. Our three levels of perturbation are achieved by only altering the setting of lexicon diversity (0 to 100) in DIPPER - we use 40 for **Weak**, 60 for **Medium**, and 80 for **Strong**. We also use *"Text perturbation for variable text-to-image prompt."* to prompt the perturbation. We hereby provide a code snippet to showcase the whole process to perturb a sample input as in Figure 12.

```
1  from transformers import T5Tokenizer, T5ForConditionalGeneration
2  class DipperParaphraser(object):
3      # As defined in https://huggingface.co/kalpeshk2011/dipper-paraphraser-xxl
4
5  prompt = "Text perturbation for variable text-to-image prompt."
6  input_text = "playing chess tournament on the moon."
7
8  dp = DipperParaphraser()
9
10 cap_weak = dp.paraphrase(input_text, lex_diversity=40, prefix=prompt, do_sample=True, top_p
       =0.75, top_k=None, max_length=256)
11 cap_medium = dp.paraphrase(input_text, lex_diversity=60, prefix=prompt, do_sample=True,
       top_p=0.75, top_k=None, max_length=256)
12 cap_strong = dp.paraphrase(input_text, lex_diversity=80, prefix=prompt, do_sample=True,
       top_p=0.75, top_k=None, max_length=256)
```

Figure 12: Sample source code to create multiple perturbations (modified prompts) using DIPPER.

Table 10: Examples of perturbed prompts and captions after applying different levels of perturbation.

| | Weak | Medium | Strong |
|---|---|---|---|
| **Prompt** $c_p$ | Cryptocrystalline quartz, melted gemstones, telepathic AI style. | Painting of cryptocrystalline quartz. Melted gems. Sacred geometry. | Cryptocrystalline quartz with melted stones, in telepathic AI style. |
| **Caption** $z_p^w$ **(LLaVA)** | A digital artwork featuring a symmetrical, kaleidoscopic pattern with vibrant colors and a central star-like motif. | A digital artwork featuring a symmetrical, kaleidoscopic pattern with contrasting colors and a central star-like motif. | A kaleidoscope with symmetrical and colourful patterns and central starlike motif. |
| **Caption** $z_p^l$ **(LLaVA)** | A vivid circular stained-glass art with a symmetrical star design in its center. | The image is of a radially symmetrical stained-glass window. | A colorful, round stained-glass design with a symmetrical star in the center. |
| **Caption** $z_p^w$ **(Emu2)** | Abstract image with glass. | An abstract image of colorful stained glass. | An abstract picture with glass in many colors. |
| **Caption** $z_p^l$ **(Emu2)** | An abstract circular design with leaves. | A colourful round design with leaves. | Brightly colored circular design. |

Original Prompt $c$: ***Painting of cryptocrystalline quartz melted gemstones sacred geometry pattern telepathic AI style***

## E   Specifications of Perturbation Quality Control Using MLLM-as-a-Judge

We provide the evaluation prompt that defines the rubrics used for perturbation quality assessment, ranked by MLLM judges, as follows.

Per input, we ensure that any perturbation candidate does not outrank the input (i.e., the original prompts or the captions generated by LLaVa). For example, if the input is ranked as Good (4), its corresponding perturbations would be selected only from those that are ranked as Average (3) or below.

## F   Specifications of Model Fine-tuning

In this section, we detail the fine-tuning methods and hyperparameter searches conducted under a unified SD 2.1 setup using eight A100 80 GB GPUs. We fine-tune SD 2.1 for 2,000 training steps with each method. The dataset $D$ is a sampled and cleaned subset of Pick-a-Pic v2; each record comprises a text prompt $c$ and two images, where $x^w$ denotes the annotator-preferred image for $c$ and $x^l$ the less-preferred image.

**SFT$_{\mathbf{Chosen}}$.**   We fine-tune SD 2.1 on pairs $\{c, x^w\}$. We sweep batch size in $\{64, 128\}$ and learning rate in $\{10^{-10}, 10^{-9}, 10^{-8}, 10^{-7}\}$. The strongest configuration uses batch size 64 with learning rate $10^{-9}$. Larger learning rates substantially degrade quality, and increasing the batch size to 128 at fixed learning rate $10^{-9}$ reduces overall performance.

---

**Evaluation Prompt for Assessing Perturbation (Modified Prompt) Quality Using an MLLM Judge**

You are given an image, and a text which is an image caption generated by another AI model for the given image. Rate how well the caption describes the image on a discrete scale from 1 to 5. Here is the detailed scoring rubric for evaluating the quality of the caption:

Excellent (5): the caption perfectly adheres to the image, excelling in relevance, accuracy, comprehensiveness, creativity, and granularity. It provides an insightful, detailed, and thorough answer, indicating a deep and nuanced understanding of the image.

Good (4): the caption is well-aligned with the image, demonstrating a high degree of relevance, accuracy, and comprehensiveness. It shows creativity and a nuanced understanding of the topic, with detailed granularity that enhances the response quality.

Average (3): the caption adequately addresses the image, showing a fair level of relevance, accuracy, and comprehensiveness. It reflects a basic level of creativity and granularity but may lack sophistication or depth in fully capturing the image.

Fair (2): the caption addresses the user's instruction partially, with evident shortcomings in relevance, accuracy, or comprehensiveness. It lacks depth in creativity and granularity, indicating a superficial understanding of the image.

Poor (1): the caption significantly deviates from the image and fails to address the query effectively. It shows a lack of relevance, accuracy, and comprehensiveness. Creativity and granularity are absent or poorly executed.

---

**Diffusion-DPO.** We train SD 2.1 on triples $\{c, x^w, x^l\}$ with batch size fixed to 128 and learning rate fixed to $10^{-8}$. We sweep the regularization coefficient $\beta$ over $\{500, 1000, 2000, 2500, 5000\}$. Performance peaks at a moderate $\beta$ around 2000, with small but consistent declines away from this setting, which aligns with the expected trade-off governed by $\beta$.

**MaPO.** We also use triples $\{c, x^w, x^l\}$ with batch size fixed to 128 to fine-tune SD 2.1. Guided by the design of the method, we focus the search on the small-$\beta$ regime with $\beta \in \{0.01, 0.1\}$ and learning rate in $\{10^{-8}, 10^{-7}\}$. The most favorable setting is $\beta = 0.01$ with learning rate $10^{-8}$. Increasing $\beta$ or the learning rate degrades performance, reflecting sensitivity to heavier margin weighting and aggressive steps.

**DCPO.** We explore two families of configurations that differ only in how captions and perturbations are constructed while keeping the SD 2.1 backbone and compute budget unchanged. DCPO-c and DCPO-p fine-tune SD 2.1 with the DCPO objective using captions produced by LLaVA and Emu2, respectively, and evaluate three perturbation levels. Unless otherwise noted, the main results for DCPO-p use a weak perturbation applied to the original prompt, and Table 3 reports DCPO-p performance across the other perturbation levels. DCPO-h uses distinct captions for each image; for every training instance we form two caption–image pairs, $(c^w, x^w)$ and $(c^l, x^l)$, where $c^w$ describes the preferred image and $c^l$ describes the less-preferred image. Captions are generated with LLaVA and then perturbed according to the DCPO-h protocol. For DCPO and Diffusion-DPO we sweep $\beta$ over $\{500, 1000, 2000, 2500, 5000\}$ with batch size 128 and learning rate $10^{-8}$. Within this range DCPO attains its best overall performance at $\beta = 5000$, and the reported DCPO-h results correspond to a medium perturbation applied to $c^l$. Table 3 presents DCPO-h performance across additional perturbation levels, including experiments that perturb $c^w$, and Table 11 reports DCPO-h results when captions are generated by Emu2. The key findings indicate that perturbations applied to short captions do not improve performance and often produce worse outcomes compared to DCPO-c (Emu2).

We also examine sensitivity to distribution shift by constructing in-distribution and out-of-distribution splits using LLaVA and Emu2 within the captioning setup. As shown in Figure 13, in-distribution data generally outperforms out-of-distribution data, while the most significant improvement is observed with the hybrid configuration reported in Figure 6.

Finally, Table 12 reports the $\beta$ sweep for DCPO-h under the medium perturbation setting alongside Diffusion-DPO. The results highlight that, across a range of $\beta$ values, DCPO-h can outperform Diffusion-DPO, underscoring the effectiveness of the DCPO framework. Based on these observations, we recommend validating $\beta$ for each dataset and task, since the preferred setting can shift with distributional differences and metric emphasis.

Table 11: Results of the perturbation method applied to Emu2 captions across different levels.

| Method | Pair Caption | Perturbed Level | Pickscore (↑) | HPSv2.1 (↑) | ImageReward (↑) | CLIPscore (↑) | GenEval (↑) |
|---|---|---|---|---|---|---|---|
| DCPO-h | $(z^w, z_p^w)$ | weak | 20.10 | 21.23 | 49.7 | 26.87 | 0.5003 |
| DCPO-h | $(z^w, z_p^l)$ | weak | 20.32 | 23.4 | 53.8 | 27.06 | 0.5070 |
| DCPO-h | $(z^w, z_p^w)$ | medium | 20.31 | 23.08 | 53.2 | 27.01 | 0.4895 |
| DCPO-h | $(z^w, z_p^l)$ | medium | 20.33 | 23.22 | 53.8 | 27.09 | 0.5009 |
| DCPO-h | $(z^w, z_p^w)$ | strong | 20.31 | 22.95 | 53.1 | 27.11 | 0.4878 |
| DCPO-h | $(z^w, z_p^l)$ | strong | 20.35 | 23.24 | 53.63 | 27.08 | 0.5050 |

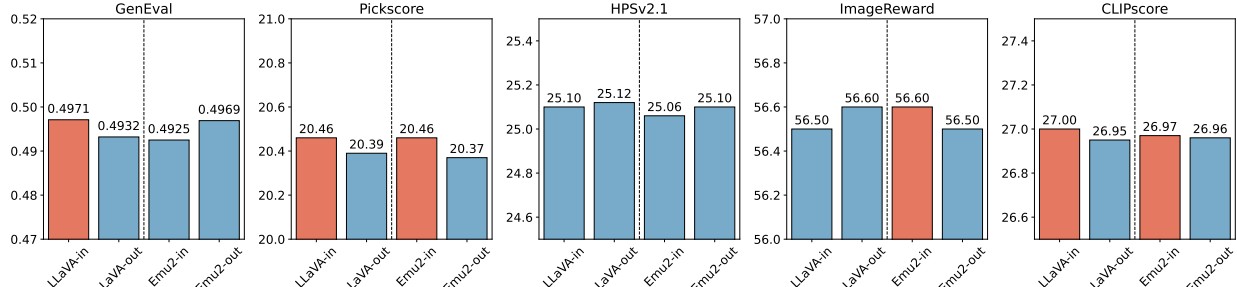

Figure 13: Comparison of DCPO-c performance on in-distribution and out-of-distribution data.

Table 12: Results of DCPO-h across different $\beta$.

| Method | $\beta$ | Pickscore (↑) | HPSv2.1 (↑) | ImageReward (↑) | CLIPscore (↑) | GenEval (↑) |
|---|---|---|---|---|---|---|
| DCPO-h | 500 | 20.43 | **26.42** | 58.10 | 27.02 | **0.5208** |
| Diffusion-DPO | 500 | 20.32 | 24.79 | 54.22 | 26.83 | 0.4926 |
| DCPO-h | 1000 | 20.51 | 26.12 | 58.2 | 27.10 | 0.4900 |
| Diffusion-DPO | 1000 | 20.34 | 24.99 | 55.06 | 26.86 | 0.4935 |
| DCPO-h | 2000 | **20.61** | 25.83 | **58.36** | **27.19** | 0.5056 |
| Diffusion-DPO | 2000 | 20.36 | 25.10 | 56.43 | 26.98 | 0.4858 |
| DCPO-h | 2500 | 20.53 | 25.81 | 58.00 | 27.02 | 0.5036 |
| Diffusion-DPO | 2500 | 20.35 | 25.11 | 55.24 | 26.86 | 0.4920 |
| DCPO-h | 5000 | 20.57 | 25.62 | 58.2 | 27.13 | 0.5100 |
| Diffusion-DPO | 5000 | 20.34 | 25.16 | 55.49 | 26.87 | 0.4974 |

# G  GPT-4o as Evaluator

**Positional Bias.** To obtain binary preferences from the API evaluator, we follow the approach outlined in the MaPO paper (Hong et al., 2024). To address positional bias in GPT-4o's evaluations, we alternate the positions of the images across different criteria on 100 samples (explained in Section 4). The results in Table

13 show that DCPO consistently achieves better performance than Diffusion-DPO, even when positional bias is accounted for.

Table 13: Comparison of DCPO and Diffusion-DPO across different positions in prompt of GPT-4o based on various criteria.

| Method | General Preference | Visual Appeal | Prompt Alignment |
|---|---|---|---|
| DCPO-h | **58%** | **64.5%** | **56.5%** |
| Diffusion-DPO | 42% | 35.5% | 43.5% |

**Human Study.** While recent work Hong et al. (2024) shows that GPT-4o can judge two images given the same prompt, we conducted a small human study to independently assess this alignment. We randomly sampled 100 image pairs produced by SD 2.1 fine-tuned with DCPO-h and Diffusion-DPO, and recruited three volunteer researchers to evaluate visual appeal. Each annotator answered the question *"Which image is more visually appealing?"* The comparison performance of DCPO-h and Diffusion-DPO on Visual Appeal, reported in Tables 13 and 14, indicates that GPT-4o judgments closely align with human preferences.

Table 14: Comparison of DCPO and Diffusion-DPO across different positions in the prompt of GPT-4o based on various criteria.

| Method | Volunteer 1 | Volunteer 2 | Volunteer 3 | Average |
|---|---|---|---|---|
| Win Rate of DCPO-h | 64% | 66% | 69% | **66.33%** |
| Win Rate of Diffusion-DPO | 36% | 34% | 41% | 33.67% |

**Judgment Prompts.** Similarly to Diffusion-DPO, we use three distinct questions to evaluate the images generated by DCPO-h and other baseline models, all utilizing SD 2.1 as the backbone. These questions are presented to GPT-4o to identify the preferred image. In the following, we provide details of the prompts used.

---

**GPT-4o Evaluation Prompt for Q1: General Preference**

Select the output (a) or (b) that best matches the given prompt. Choose your preferred output, which can be subjective. Your answer should ONLY contain: Output (a) or Output (b).

## Prompt:
{prompt}

## Output (a):
The first image attached.

## Output (b):
The second image attached.

## Which image do you prefer given the prompt?

---

---

**GPT-4o Evaluation Prompt for Q2: Visual Appeal**

Select the output (a) or (b) that best matches the given prompt. Choose your preferred output, which can be subjective. Your answer should ONLY contain: Output (a) or Output (b).

## Prompt:
{prompt}

## Output (a):
The first image attached.

## Output (b):
The second image attached.

## Which image is more visually appealing?

---

**GPT-4o Evaluation Prompt for Q3: Prompt Alignment**

Select the output (a) or (b) that best matches the given prompt. Choose your preferred output, which can be subjective. Your answer should ONLY contain: Output (a) or Output (b).

## Prompt:
{prompt}

## Output (a):
The first image attached.

## Output (b):
The second image attached.

## Which image better fits the text description?

---

# H    Additional Generated Samples

We also present additional samples for qualitative comparison generated by SD 2.1, $SFT_{Chosen}$, Diffusion-DPO, MaPO, and DCPO-h from prompts on Pickscore, HPSv2, and GenEval benchmarks.

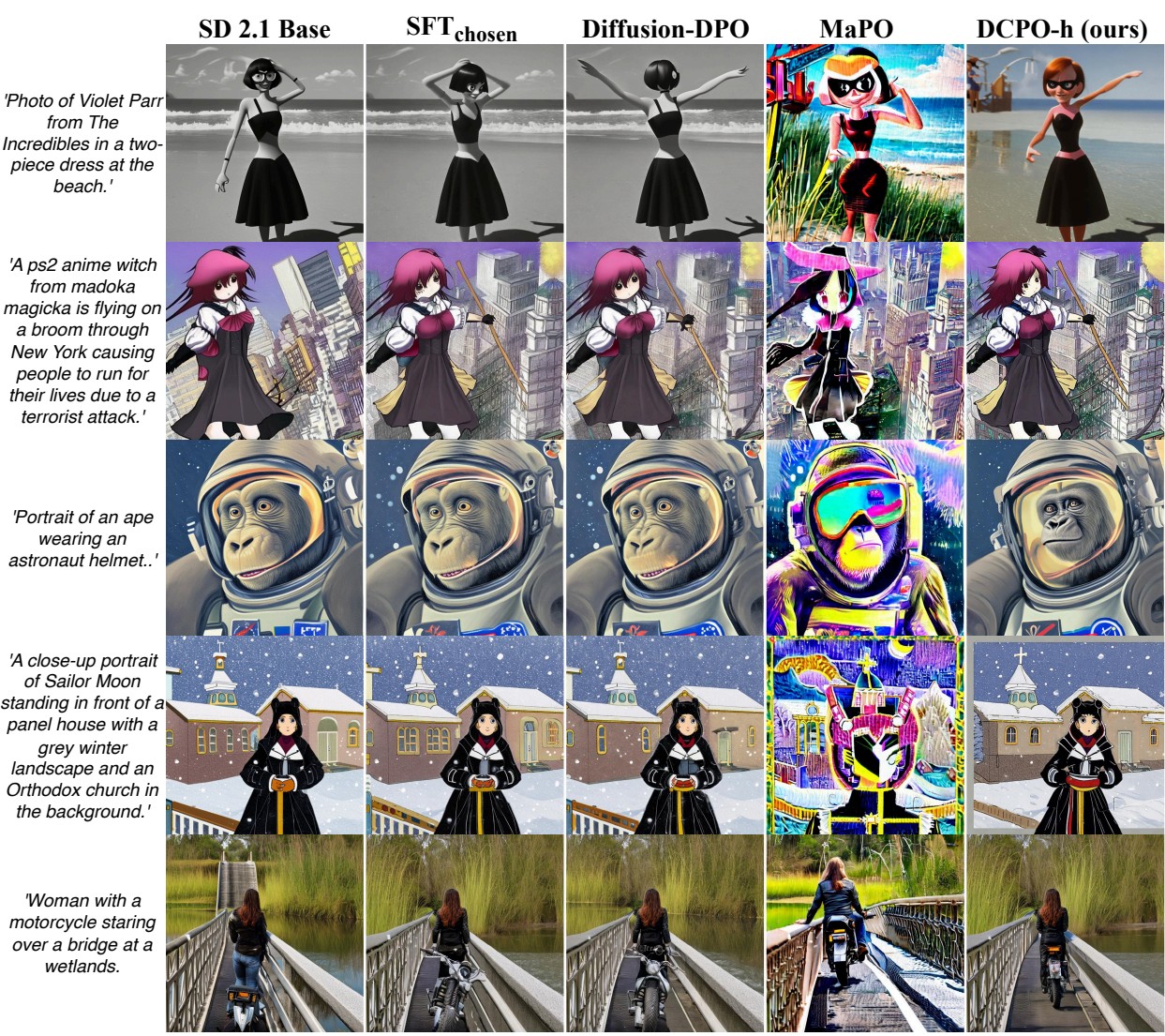

Figure 14: Additional generated outcomes using prompts from HPSv2 benchmark.

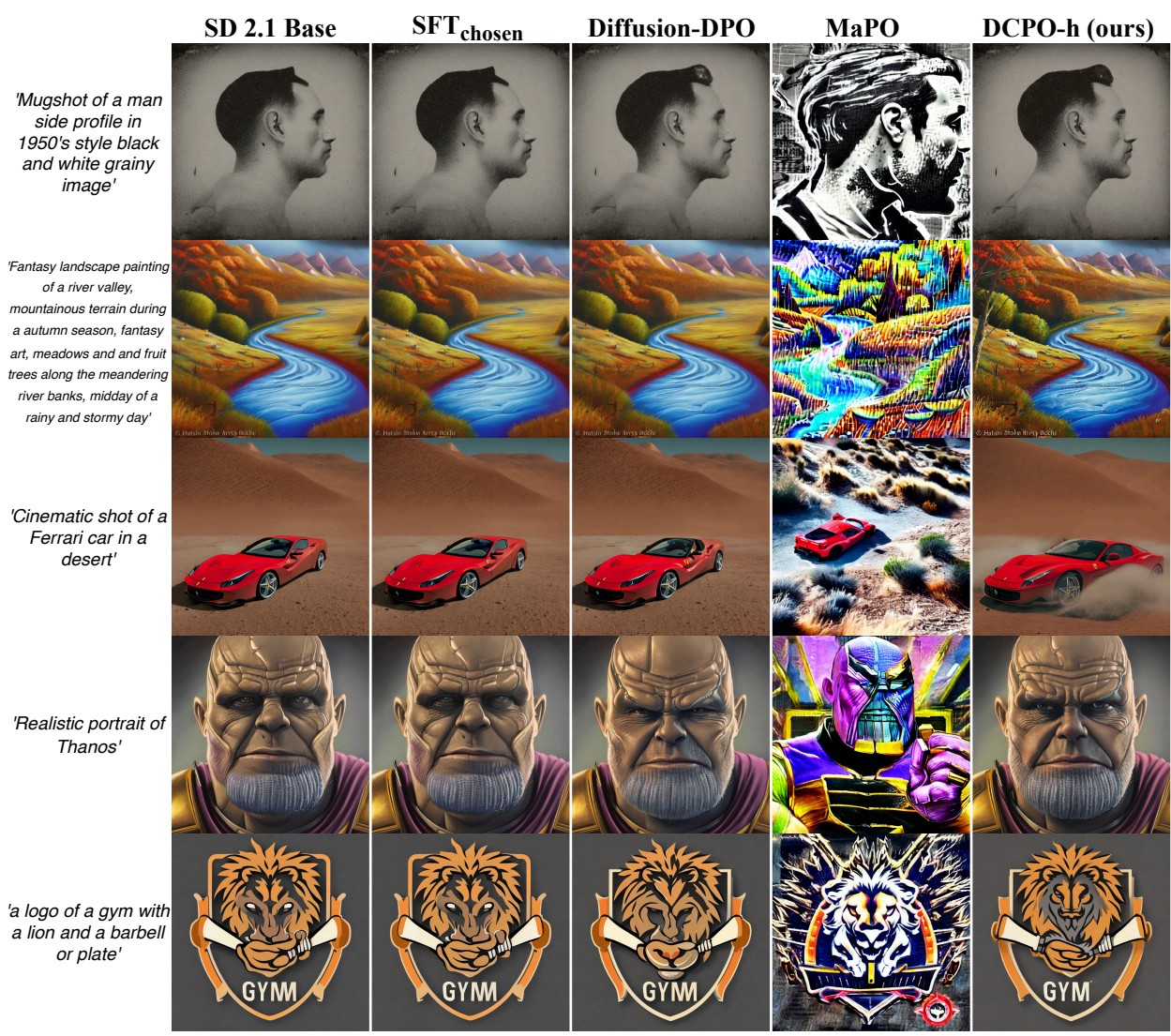

Figure 15: Additional generated outcomes using prompts from Pickscore benchmark.

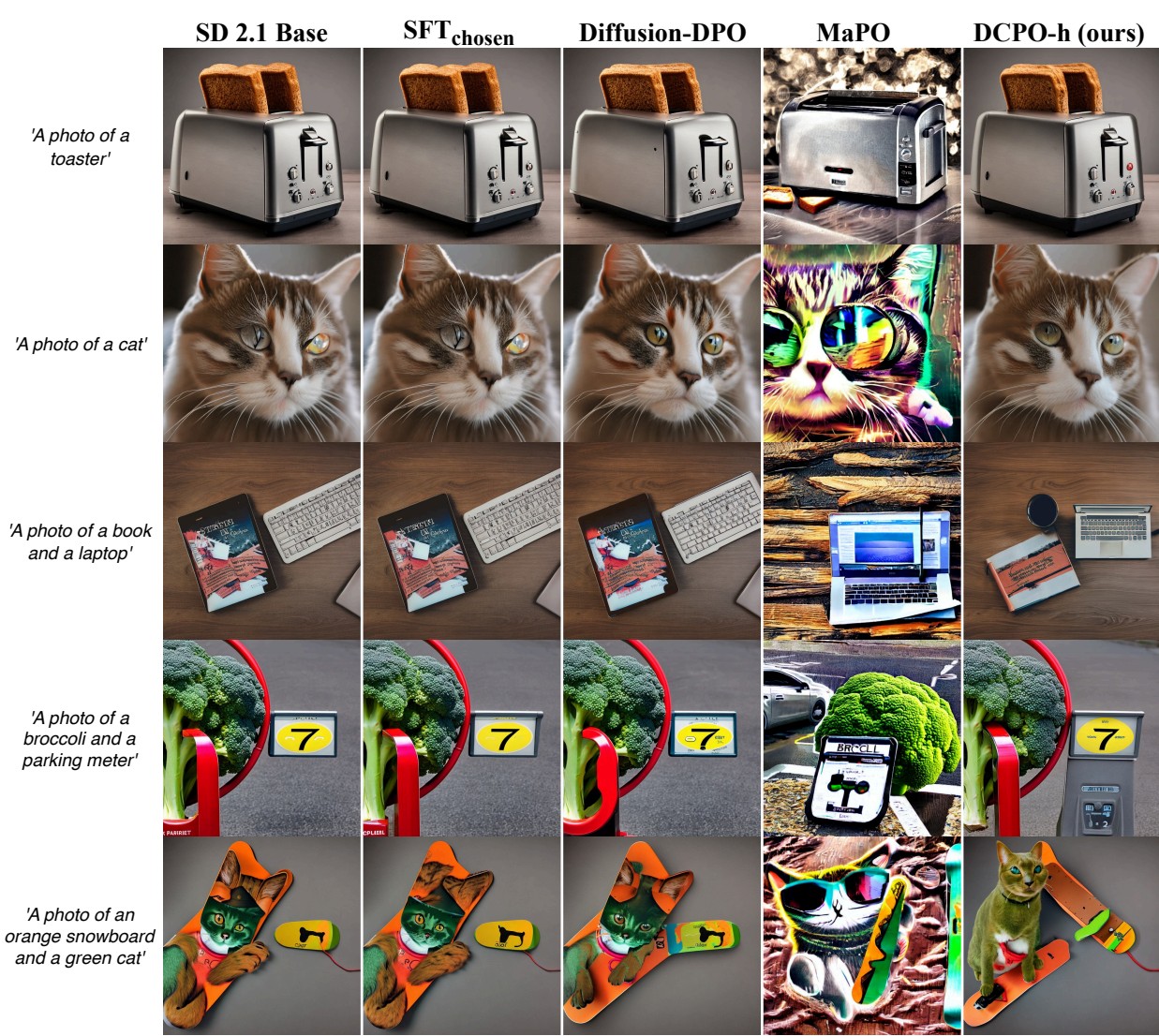

Figure 16: Additional generated outcomes using prompts from GenEval benchmark.

