# OpenReview forum: "Dual Caption Preference Optimization for Diffusion Models"
_TMLR — Accepted by TMLR_

### Review · Reviewer_w5K7 · 2025-06-16

**Summary Of Contributions:**

The paper studies the variant of direct alignment algorithms in the text-to-image diffusion models. The paper proposes modifying the prompts to ensure the distributional gap between the chosen and rejected contents, resulting in the DCPO methods. Multiple alignment metrics and previous methods were employed to demonstrate the empirical validity of DCPO. The paper's insights highlight the importance of carefully considering the quality of the prompt-image alignment for the better results in the downstream alignment performance of the diffusion models.

**Audience:**

Yes

**Broader Impact Concerns:**

No other concerns on the ethical implications of the work.

**Claims And Evidence:**

No

**Requested Changes:**

Detailed requests for changes are discussed above. All three weaknesses should be addressed. Especially, the first weakness should be addressed to strengthen the theoretical foundation of the proposed method, and the third weakness should be addressed to strengthen the empirical justification that DCPO outperforms the baseline methods. Without addressing the two weaknesses, the DCPO lacks justification of its necessity both theoretically and empirically.

**Strengths And Weaknesses:**

### **Strengths of the paper**

1. The paper adopts diverse baseline methods to present DCPO's empirical strength, including Diffusion-DPO, MaPO, and Diffusion-KTO.

2. Diverse alternatives for modifying the prompts of binary preference dataset are explored. This provides a better empirical understanding that the prompts' contextual diversity could be crucial for T2I diffusion model alignment.


### **Questions and Weaknesses of the paper**

1. **Is "Conflict Distribution" truly a problem?** - While the method is built on the *conflict distribution* as a core issue in the existing pairwise preference datasets for the T2I diffusion models, it is not clearly justified.
    - First, the concept is not clearly defined in the paper. The use of arbitrary metrics like CLIPScore (Figure 3) and VQAScore (Figure 9) confuses what exactly the conflict distribution is referring to. The authors mention *"Ideally, the distributions of these responses
should differ significantly"*, but CLIPScore and VQAScore are not measuring the difference between $y^w$ and $y^l$, they measure the alignment between the prompt and the corresponding image. This further makes the concept of *conflict distribution* unclear.
    - Second, chosen and rejected images both being similarly aligned to the prompt is not an issue. The early attempts in the preference alignment leveraged pairwise datasets that comprise multiple models (which is not $p_\mathrm{ref}$) that generate responses. However, it is shown that using the responses (images) generated from $p_\mathrm{ref}$ led to a better alignment both empirically [1] and theoretically based on the derivation of DPO [2]. Thus, the CLIPScore/VQAScore of $(x^{(i)}, c)$ for $x^{(i)} \sim p_\mathrm{ref}(\cdot|c)$ being invariant between the chosen and rejected sets is a natural phenomenon. Again, the authors are relying on *"Ideally, the distributions of these responses should differ significantly"* to justify that this is problematic, but the observations in the Figures 3 and 9 are entailed from curating the preference datasets with $(x^{(i)}, c)$ for $x^{(i)} \sim p_\mathrm{ref}(\cdot|c)$. While the empirical performance increase with different prompts could be one evidence (which is similarly reported in LLM-based reward modeling context [5]), it does not justify that the conflict distribution is problematic. The authors need to be more specific about why this is problematic.

2. **Correlation between the prompt modifier's strength and the downstream results** - As the caption/perturbed prompt generator (prompt generator from now on) is the key component that differentiates DCPO from DPO, the reader would expect a fine-grained study on the strength of the prompt generator. The paper uses LLaVa-1.6-34B and EMU2 as a caption generators and T5-XXL based model for perturbation. However, there are a lot more models that have a better performance with a better steerability, e.g., more fine-grained control on the degree of perturbation, including the proprietary models [3, 4]. Plus, the choice of LLaVa and EMU2 does not have a clear reason or standard behind it. A clearer ablation on the prompt generators will further support the strength of DCPO to the practitioners.

3. **Hyperparameter tuning for compared baselines** - The paper mentions *"We fine-tuned SD 2.1 with the best hyperparameters reported in the original papers"* (Appendix E) for the training configurations for the baseline methods, including $\mathrm{SFT}_\mathrm{Chosen}$, Diffusion-DPO, and MaPO. However, both Diffusion-DPO and MaPO papers did not report the optimal hyperparameters for SD 2.1, as both papers employed SDXL or SD 1.5. As all the models have different sizes and configurations, a separate hyperparameter tuning for the baseline methods are crucial for the fair comparison. Including those baselines and the proposed method, every training configuration should be clearly reported in the paper.

&nbsp;

**References**

[1] Meng et al., 2024, "SimPO: Simple Preference Optimization with a Reference-Free Reward".

[2] Liu et al., 2023, "Statistical Rejection Sampling Improves Preference Optimization".

[3] Chen et al., 2024, "MLLM-as-a-Judge: Assessing Multimodal LLM-as-a-Judge with Vision-Language Benchmark".

[4] Yue et al., 2023, "MMMU: A Massive Multi-discipline Multimodal Understanding and Reasoning Benchmark for Expert AGI".

[5] Sun et al., 2025, "Rethinking Reward Modeling in Preference-based Large Language Model Alignment".

---

> ### Author Response · Authors · 2025-07-29
> **Our Responses Part 4/4**
>
> > Q3. Hyperparameter tuning for compared baselines ...
>
> Thank you for the insightful comment. We have conducted comprehensive hyperparameter tuning to ensure a fair comparison across all methods. Below, we provide a summary of outcomes from our DCPO and other baseline models with various applicable hyperparameter configurations. By default, all results are obtained using Stable Diffusion 2.1 as the backbone model, a batch size of 128, and a learning rate of 1e-8, unless specified.
>
> Our proposed DCPO-h baseline is the one using Beta=5000.
>
> | Method| Beta | Pickscore | GenEval | HPSv2.1 | ImageReward | CLIPscore |
> | :---- | :---- | ----: | ----: | ----: | ----: | ----: |
> | DCPO-h | 500 |  20.43 | 0.5209 | 26.42 | 58.17 | 27.03 |
> | DCPO-h | 1000 | 20.51 | 0.4900 | 26.12 | 58.20 | 27.11 |
> | DCPO-h | 2000 | 20.62 | 0.5056 | 25.83 | 58.36 | 27.20 |
> | DCPO-h | 2500 | 20.54 | 0.5036 | 25.81 | 57.95 | 27.02 |
> | DCPO-h | 5000 | **20.57** | **0.5100** | **25.62** | **58.20** | **27.13** |
>
> For DPO, all learning rates are 1e-8.
>
> | Method | Beta | Pickscore | GenEval | HPSv2.1 | ImageReward | CLIPscore |
> | :---- | :---- | ----: | ----: | ----: | ----: | ----: |
> | DPO | 500 | 20.32 | 0.49268 | 24.79 | 54.22 | 26.83 |
> | DPO | 1000 | 20.34 | 0.49358 | 24.99 | 55.06 | 26.86 |
> | DPO | 2000 | 20.36 | 0.48577 | 25.10 | 56.43 | 26.98 |
> | DPO | 2500 | 20.35 | 0.49201 | 25.11 | 55.24 | 26.86 |
> | DPO | 5000 | 20.34 | 0.49744 | 25.16 | 55.49 | 26.87 |
>
> SFT has its best performance when using a batch size of 64 and a learning rate of 1e-9.
>
> | Method | Batch size | Learning Rate | Pickscore | GenEval | HPSv2.1 | ImageReward | CLIPscore |
> | :---- | :---- | :---- | ----: | ----: | ----: | ----: | ----: |
> | SFT | 64 | 1e-10 | 20.35 | 0.4775 | 24.79 | 54.77 | 26.90 |
> | SFT | 64 |1e-9 | 20.36 | 0.4797 | 25.09 | 56.47 | 26.99 |
> | SFT | 64 |1e-8 | 17.52 | 0.0039 | 4.36 | 2.26 | 12.70 |
> | SFT | 64 |1e-7 | 17.36 | 0.0063 | 4.58 | 1.40 | 12.47 |
> | SFT | 128 |1e-9 | 17.49 | 0.0427 | 5.45 | 2.46 | 14.19 |
> | SFT | 128 |1e-9 | 17.59 | 0.0265 | 4.66 | 1.36 | 13.07 |
>
> MaPO uses a way smaller Beta value compared to DCPO or DPO for its best performances.
>
> | Method | Beta| Learning Rate | Pickscore | GenEval | HPSv2.1 | ImageReward | CLIPscore |
> | :---- | :---- | :---- | ----: | ----: | ----: | ----: | ----: |
> | MaPO | 0.1 | 1e-8 | 20.32 | 0.4922 | 25.20 | 55.38 | 26.81 |
> | MaPO | 0.01 | 1e-8 | 20.31 | 0.4938 | 25.31 | 55.63 | 26.78 |
> | MaPO | 0.1 | 1e-7 | 19.41 | 0.4106 | 24.47 | 50.45 | 24.82 |
>
> The overall outcomes demonstrate that DCPO consistently outperforms all other methods when compared using their respective best-tuned hyperparameters.

---

> ### Author Response · Authors · 2025-07-29
> **Our Responses Part 3/4**
>
> > Q2 (continuted): Correlation between the prompt modifier's strength and the downstream results ...
>
> Below is a summary of benchmarking results under the shared Stable Diffusion 2.1 and DCPO-h \+ LLaVa captions setting, this time using different perturbation methods. We experimented with the following MLLM perturbation judge models \- Qwen Chat, Qwen2.5-VL-7B, LlaVa1.6-7B, and LlaVa1.6-13B. We also present the results from our original CLIPScore-as-a-Judge method as the baseline. For clarity, we show the averages across all perturbation strengths per perturbation judge and benchmark.
>
> | Perturb. Judge | Pickscore | GenEval | HPSv2.1 | ImageReward | CLIPScore |
> | :---- | ----: | ----: | ----: | ----: | ----: |
> | CLIPScore (ours) | **20.58**	±0.01 | **0.5051**	±0.01 | **25.58**    ±0.15 | **58.07** ±0.17 | **27.16** ±0.03 |
> | Qwen Chat | 20.46	±0.03 | 0.4911	±0.01 | 24.74	±0.07 | 55.48	±0.46 | 26.97	±0.03 |
> | Qwen2.5-VL-7B | 20.44	±0.03 | 0.4910	±0.01 | 24.62	±0.16 | 55.18	±0.14 | 26.93	±0.04 |
> | LLaVa1.6-7B | 20.44	±0.03 | 0.4911	±0.02 | 24.64	±0.08 | 55.07	±0.04 | 26.93	±0.05 |
> | LLaVa1.6-13B | 20.44	±0.02 | 0.4913	±0.01 | 24.65	±0.08 | 55.38	±0.31 | 26.93	±0.03 |
>
> The results show that all MLLM-as-a-Judge variants consistently underperform compared to our CLIPScore-based baselines across all five downstream benchmarks, regardless of the judge model’s scale or the granularity of its perturbation strengths. This further validates that our original method identifies more effective perturbations (i.e., modified prompts) that lead to improved downstream performance across multiple benchmarks. We will include a detailed account of this ablation study, along with additional clarifications on our original perturbation method, in the final version of the paper.
>
> \[5\] https://arxiv.org/abs/2303.13408
> \[6\] https://arxiv.org/abs/2402.04788

---

> ### Author Response · Authors · 2025-07-29
> **Our Responses Part 2/4**
>
> > Q2: Correlation between the prompt modifier's strength and the downstream results ...
>
> Thank you for highlighting the importance of finer-grained control over the strength of perturbation. We would like to take this chance to further clarify the intuitions behind our perturbation setup.
>
> 1. **Our original prompt modifier is highly customizable by design in terms of perturbation strength.** The modifier that we used, the T5-based DIPPER model \[5\], is already equipped with flexible steerability for textual perturbation. DIPPER supports adjustable hyperparameters that control both lexical diversity and word order variation. To construct a diverse pool of perturbed caption candidates for each input, we perform a hyperparameter sweep over DIPPER’s controllable dimensions.
> 2. **CLIPScore-as-a-Judge for selecting perturbations for PickDouble.** We then employ CLIPScore as a selection mechanism for the perturbations used in our PickDouble dataset. For each input, we select the perturbed captions that are well-separated along the CLIPScore scale from the aforementioned pool of candidates, ensuring a clear semantic gradient, while always preserving the original input as the one with the highest image-text alignment (i.e., CLIPScore affinity). The resulting levels—Weak, Medium, and Strong—reflect progressively degraded semantic fidelity relative to each input’s associated image, shown as the shifting distributions in Figure 4\.
>
> In response to your suggestion, we further conducted a series of controlled ablation studies using perturbations with finer-grained strength levels, this time ranked by MLLM judges. Below, we provide the evaluation prompt that defines the rubrics used for perturbation quality assessment, adapted from the very prompt used in the MLLM-as-a-Judge paper \[6\].
>
>     "You are given an image, and a text which is an image caption generated by another AI model for the given image. Rate how well the caption describes the image on a discrete scale from 1 to 5\. Here is the detailed scoring rubric for evaluating the quality of the caption:
>
>     Excellent (5): the caption perfectly adheres to the image, excelling in relevance, accuracy, comprehensiveness, creativity, and granularity. It provides an insightful, detailed, and thorough answer, indicating a deep and nuanced understanding of the image.
>
>     Good (4): the caption is well-aligned with the image, demonstrating a high degree of relevance, accuracy, and comprehensiveness. It shows creativity and a nuanced understanding of the topic, with detailed granularity that enhances the response quality.
>
>     Average (3): the caption adequately addresses the image, showing a fair level of relevance, accuracy, and comprehensiveness. It reflects a basic level of creativity and granularity but may lack sophistication or depth in fully capturing the image.
>
>     Fair (2): the caption addresses the user's instruction partially, with evident shortcomings in relevance, accuracy, or comprehensiveness. It lacks depth in creativity and granularity, indicating a superficial understanding of the image.
>
>     Poor (1): the caption significantly deviates from the image and fails to address the query effectively. It shows a lack of relevance, accuracy, and comprehensiveness. Creativity and granularity are absent or poorly executed."
>
> Per example, we made sure that any perturbation candidate does not outrank the original input (i.e. captions generated by LLaVa). For instance, if the original caption is ranked as Good (4), its corresponding perturbations would be selected only from those that are ranked Average (3) or below.

---

> ### Author Response · Authors · 2025-07-29
> **Our Responses Part 1/4**
>
> We are beyond grateful for your meticulous reviews. Please find our responses detailed below that address each of your concerns.
>
> > Q1: Is "Conflict Distribution" truly a problem?
>
> > Q1(a): First, the concept is not clearly defined in the paper. The use of arbitrary metrics like CLIPScore (Figure 3) and VQAScore (Figure 9) confuses what exactly the conflict distribution is referring to....
>
> Thank you for the insightful question. As discussed in Section 3.1, preference optimization was originally introduced in the context of language models. The statement “Ideally, the distributions of these responses should differ significantly” refers specifically to preference optimization in language generation. Intuitively, if the preferred and unpreferred responses are too similar for a given prompt, then log P($y_w$ | x) is approximately equal to log P($y_l$ | x), which results in a near-zero loss during optimization. In such cases, the model receives little to no learning signal and fails to improve. Prior works have shown that high similarity between response pairs for the same prompt can significantly degrade model performance.
>
> In language models, the learning objective is driven by the contrast between the preferred and unpreferred responses. The model learns to increase the likelihood of the preferred response relative to the unpreferred one. While diffusion models operate differently, learning through a forward and reverse denoising process conditioned on a prompt, we are inspired by the same core idea.
>
> To adapt this concept to image generation, we analyzed the semantic similarity between the prompt and both the preferred and unpreferred images in the current preference dataset. Specifically, we used CLIPScore and VQAScore to measure alignment \[3,4\]. Our findings indicate substantial overlap in the distributions of these scores for preferred and unpreferred images, which we refer to as "conflict." This overlap reflects a lack of clear separation in semantic alignment, which can hinder learning during preference optimization.
>
> Throughout the paper, we emphasize that our focus in the visual domain is on semantic distributions, as outlined in Hypothesis 1\. Nonetheless, we will revise the relevant explanation in the manuscript to make this point clearer.
>
> \[1\] https://arxiv.org/pdf/2502.14560v2
> \[2\] https://aclanthology.org/2025.findings-acl.282.pdf
> \[3\] https://arxiv.org/pdf/2305.13308
> \[4\] https://arxiv.org/pdf/2307.06350v1
>
> > Q1(b). Second, chosen and rejected images both being similarly aligned to the prompt is not an issue. The early attempts in the preference alignment leveraged pairwise datasets that comprise multiple models (which is not ) that generate responses...
>
> Thank you to the reviewer for pointing out the relevant works. In the RSO method, the reference model is a supervised fine-tuned checkpoint, and in SimPO, it is based on both supervised and instruction fine-tuned models. These fine-tuned models tend to produce more consistent and aligned preferences, which supports their optimization process.
>
> However, in the text-to-image domain, the distinction between pre-trained and supervised fine-tuned models is less pronounced. We argue that current text-to-image models are predominantly pre-trained, and to improve alignment performance, preference optimization alone may be sufficient without the need for supervised fine-tuning. This marks a key difference between research in language and vision domains.
>
> We also acknowledge that Pic-a-Pic v2, the dataset used in our study, is a preference dataset where images are generated by various models, including members of the Stable Diffusion family and Dreamlike Photoreal 2.0. As a result, we observe that the alignment scores (CLIPScore and VQAScore) for preferred and unpreferred images often exhibit high similarity, making them less informative for effective preference optimization.
>
> To clarify, we do not claim that existing methods degrade alignment performance. Rather, we believe that new approaches such as DCPO offer a promising path forward by pushing text-to-image models to achieve stronger alignment and overall generation quality.

---

> ### Author Response · Authors · 2025-07-29
> **Our responses are in reverse order.**
>
> Please find our multi-part responses following the bottom-up order as we are unable to edit any submitted comments. We apologize for any inconvenience this might cause.

---

### Review · Reviewer_HcS5 · 2025-06-22

**Summary Of Contributions:**

This paper introduces Dual Caption Preference Optimization (DCPO), a framework for aligning text-to-image diffusion models with human preferences, to address two major issues in existing preference optimization approaches:

1. Conflict distribution. Preferred and less-preferred images often share too much semantic overlap under the same prompt.

2. Irrelevant prompts. The same prompt can be poorly aligned with the less-preferred image, reducing optimization effectiveness.

It also introduces a new dataset, Pick-Double Caption, derived from Pick-a-Pic v2, which contains dual captions to reduce semantic overlap and improve alignment performance.

**Audience:**

No

**Broader Impact Concerns:**

N/A. Ethical Consideration is discussed on page 12.

**Claims And Evidence:**

Yes

**Requested Changes:**

1. Comparisons with more baselines.

2. Justification of computational overhead.

3. (not necessary) Human Evaluation.

**Strengths And Weaknesses:**

Strengths.

1. Theoretical Motivation and Hypothesis-Driven Evaluation. The paper is not only empirical but also theory-informed.

---

Weaknesses.

1. Computational Overhead in Caption Generation. A key limitation of DCPO lies in the computational cost and scalability of its pipeline. As stated in the Limitations section, both the captioning (DCPO-c) and perturbation (DCPO-p/h) methods require significant computational resources.

2. Limited baselines. Several related baselines are missed in comparison, e.g., SePPO [1].

3. Limited Generalization Beyond Pick-a-Pic v2. While the authors introduce the Pick-Double Caption dataset in Section 4.1, it is derived solely from Pick-a-Pic v2, limiting diversity in training data sources.

4. No Human Evaluation Beyond GPT-4o. While the paper includes GPT-4o-based judgments in Figure 7, no direct human evaluation, e.g., crowdsourced pairwise comparisons, is conducted.

[1] Zhang, Daoan, et al. "SePPO: Semi-Policy Preference Optimization for Diffusion Alignment." arXiv preprint arXiv:2410.05255 (2024).

---

> ### Author Response · Authors · 2025-07-29
> **Our Responses**
>
> Thank you so much for your thoughtful suggestions and comments. Please find our clarifications detailed below for each of your concerns.
>
> > Computational Overhead in Caption Generation. A key limitation of DCPO lies in the computational cost and scalability of its pipeline. As stated in the Limitations section, both the captioning (DCPO-c) and perturbation (DCPO-p/h) methods require significant computational resources.
>
> Thank you for the insightful comment. We have elaborated on the cost and scalability aspects of our pipeline in Appendix C. While we acknowledge that the captioning and perturbation steps introduce additional computational overhead, these operations are performed only once during the preprocessing stage and do not impact training or inference time.
>
> To offer flexibility in computational requirements, we introduce three variants of DCPO: DCPO-c, DCPO-p, and DCPO-h, each with different preprocessing needs. For example, DCPO-c and DCPO-h involve generating captions, whereas DCPO-p is more lightweight, requiring only perturbation of the less-preferred caption.
>
> In practice, the captioning step is highly scalable. Generating captions for 20,000 images using LLaVA takes less than 12 hours on a single A100 80GB GPU. This process can be significantly accelerated with multi-GPU setups, making it a practical and scalable component of the pipeline.
>
> > Limited baselines. Several related baselines are missed in comparison, e.g., SePPO.
>
> Thank you for the suggestion. In response to your comment, we have included a comparison between SePPO, SPO, and DCPO using multiple evaluation metrics.
>
> We would like to clarify that DCPO is designed as an offline, one-step optimization algorithm. It does not require generating both good and bad images during the optimization process. In contrast, SePPO is an iterative method that relies on generating bad images during training, which requires maintaining and sampling from different checkpoints of the diffusion model. In our observations, these bad images are not generated in advance but are produced dynamically during optimization, introducing additional complexity and computational overhead.
>
>
> | Method | Pickscore | GenEval | HPSv2.1 | ImageReward | CLIPscore |
> | :---- | ----: | ----: | ----: | ----: | ----: |
> | SePPO-lr-1e8 | 20.34 | 0.50567 | 25.09 | 55.21	 | 26.83 |
> | SePPO-lr-5-e9 | 20.32 | 0.50232 | 25.08		 | 55.24 | 26.82 |
>
> Similar to SePPO, we also ran experiments with SPO \[1\], which is an online algorithm, and included a comparison with DCPO in the table below. The results show that DCPO consistently outperforms SPO across all benchmarks.
>
> | Method | Pickscore | GenEval | HPSv2.1 | ImageReward | CLIPscore |
> | :---- | ----: | ----: | ----: | ----: | ----: |
> | SPO | 19.30 | 0.4449 | 24.40 | 46.45 | 25.91 |
> | DCPO-c | 20.46 | 0.4971 | 25.10 | 56.60 | 27.00 |
> | DCPO-h | 20.57 | 0.5100 | 25.62 | 58.20 | 27.13 |
> | DCPO-p | 20.28 | 0.4906 | 25.42 | 54.20 | 26.98 |
>
> \[1\] https://arxiv.org/pdf/2406.04314
>
> > Limited Generalization Beyond Pick-a-Pic v2. While the authors introduce the Pick-Double Caption dataset in Section 4.1, it is derived solely from Pick-a-Pic v2, limiting diversity in training data sources.
>
> Thank you for the comment. We address the generalization ability of DCPO in Appendix B.3 by applying it to a high-quality, independent preference dataset. To further demonstrate the versatility of DCPO, we fine-tune Stable Diffusion 2.1 using both Diffusion-DPO and DCPO on the Rapidata dataset \[2\]. As shown in Table 6, DCPO variants consistently outperform the Diffusion-DPO baseline across multiple evaluation metrics, including GenEval, PickScore, HPSv2.1, ImageReward, and CLIPScore.
>
> | Method | Pickscore | GenEval | HPSv2.1 | ImageReward | CLIPscore |
> | :---- | ----: | ----: | ----: | ----: | ----: |
> | Diffusion-DPO | 20.34 | 0.4813 | 25.10 | 55.4 | 26.84 |
> | DCPO-c | 20.44 | 0.4867 | 25.43 | 55.7 | 26.86 |
> | DCPO-h | 20.42 | 0.4978 | 25.10 | 55.6 | 26.91 |
>
> \[2\] [https://arxiv.org/pdf/2409.11904](https://arxiv.org/pdf/2409.11904)
>
> > No Human Evaluation Beyond GPT-4o. While the paper includes GPT-4o-based judgments in Figure 7, no direct human evaluation, e.g., crowdsourced pairwise comparisons, is conducted.
>
> Thank you for the valuable comment. To further assess the effectiveness of our approach, we randomly selected 100 image pairs generated by Stable Diffusion 2.1 fine-tuned with DCPO-h and Diffusion-DPO. We recruited three volunteer researchers to evaluate the images based on visual appeal. Each annotator was asked the question: “Which image is more visually appealing?”
>
> The results of this human evaluation are presented in the following table:
>
> | Volunteer | Win Rate of DCPO  | Win Rate of Diffusion-DPO |
> | :---- | :----: | :----: |
> | 1 | 64 | 36 |
> | 2 | 66 | 34 |
> | 3 | 69 | 31 |
> | Average | 66.33 | 33.67 |
>
> The average win rate of DCPO is 66.33%, which aligns closely with the results obtained using the GPT-4o evaluation.

---

> > ### Author Response · Authors · 2025-07-29
> > **Typo correction for the first table featuring SePPO**
> >
> > We would like to attach a corrected version of the table that features SePPO in our original response.
> >
> > | Method | Learning Rate | Pickscore | GenEval | HPSv2.1 | ImageReward | CLIPscore |
> > | :---- | :---- | ----: | ----: | ----: | ----: | ----: |
> > | SePPO | 1e-8 | 20.34 | 0.50567 | 25.09 | 55.21 | 26.83 |
> > | SePPO | 5e-9 | 20.32 | 0.50232 | 25.08 | 55.24 | 26.82 |

---

> > > ### Comment · Reviewer_HcS5 · 2025-07-29
> > > **Thanks for the responses.**
> > >
> > > Thank the authors for the responses.
> > >
> > > It is not required to compare with Diffusion-RainbowPA [3] as it is published very recently in TMLR. Compared to the results in the responses, I saw their reproduces, including SPO and SePPO, have much higher scores in Table 1-4 and 9. What is the reason of that? (As I may miss some experimental details, some textual explanation may be sufficient.)
> > >
> > > [3] Diffusion-RainbowPA: Improvements Integrated Preference Alignment for Diffusion-based Text-to-Image Generation. TMLR 2025. https://openreview.net/forum?id=KY0TSY2bx8

---

> > > > ### Author Response · Authors · 2025-07-29
> > > >
> > > > Thank you for the response and for sharing the fresh work. The main distinction between our results and those of Diffusion-RainbowPA lies in the dataset: they used a preprocessed version of Pic-a-Pick v1, while we used a subset of Pic-a-Pick v2. As noted in the SePPO paper, sampling is a crucial aspect of the method and may require up to seven iterations to identify the optimal output.
> > > >
> > > > However, neither the SePPO nor the Diffusion-RainbowPA papers clearly indicate which checkpoints their reported results correspond to. We assume that they present the best-performing checkpoints. To ensure a fair comparison with SePPO and DCPO, we restrict training of SePPO to K=2.
> > > >
> > > > With due respect to prior works such as SePPO and SPO, we believe that online learning and iterative strategies fall outside the intended scope of the DCPO framework. Our approach is therefore focused primarily on offline methods.

---

> > > > > ### Comment · Reviewer_HcS5 · 2025-08-04
> > > > > **Thanks for the explanation.**
> > > > >
> > > > > The dataset explanation makes sense.
> > > > >
> > > > > A friendly remind: It is allowed to upload the revised pdf in TMLR, which would be more clear to reflect the changes.

---

> > > > > > ### Author Response · Authors · 2025-08-05
> > > > > >
> > > > > > Thank you for the reminder. We will upload the revised version to ensure the changes are clearly reflected.

---

### Review · Reviewer_GyAx · 2025-07-17

**Summary Of Contributions:**

The authors propose a method to improve human preference alignment in diffusion models. They identify two issues in existing datasets: conflict distribution and irrelevant prompts, and address them using DCPO, which augments image pairs with distinct captions generated by VLMs and LLMs. Empirically, DCPO improves performance on several text-to-image generation benchmarks.

**Audience:**

Yes

**Broader Impact Concerns:**

Not applicable.

**Claims And Evidence:**

No

**Requested Changes:**

1. The notion of conflict distribution is not clearly justified. Specifically, why should the distributions of $y^w$ and $y^l$ differ significantly under the same prompt $c$? Since both $y^w$ and $y^l$ are sampled from $p_\theta(\cdot | c)$, one would expect considerable overlap in their distribution. It is unclear whether this overlap is inherently problematic or simply reflects the stochasticity of the generative model. Additionally, in Figure 3, it is not specified whether the authors are measuring $p_\theta(\cdot | c)$ using a single shared prompt or $p_\theta(\cdot)$ using multiple prompts.

2. Extra line space before Equation (9).

3. While the empirical benefit of using distinct captions is demonstrated, it remains unclear how introducing $z^w$ and $z^l$ improves the model in terms of the underlying data distribution or optimality. For example, how does the inclusion of dual captions change the effective training distribution or the optimization objective? A formal or conceptual explanation of the role of $z^w$ and $z^l$ in enhancing preference separation would strengthen the contribution.

**Strengths And Weaknesses:**

# Strengths
1. The paper identifies two under-explored issues in preference datasets and proposes a solution. This analysis adds novel insights into the limitations of current diffusion alignment methods.
2. The paper is clearly written and easy to follow. The figures are well-designed and effectively illustrate the proposed method.
3. The empirical results are strong, releasing a new dataset and showing consistent improvements across a wide range of benchmarks.
4. The paper introduces a simple and effective extension of DPO and explains why dual captioning improves optimization.

# Weaknesses
1. The motivation behind the conflict distribution issue could be better justified. It is not clearly shown why semantic overlap between preferred and less-preferred samples is inherently problematic.
2. It is not well explained how Equation (8) is derived or verified. Specifically, how the use of two distinct captions ($z^w$, $z^l$) directly improves the optimization process. How exactly does this modification alter the RLHF-style objective?

---

> ### Author Response · Authors · 2025-07-29
> **Our Responses**
>
> Regarding the question: *“Why should the distributions of $y_w$  and $y_l$ differ significantly under the same prompt $c$?”*, we point to recent works (e.g., \[1\]\[2\]) that show the choice and construction of the preference dataset significantly impacts the model's performance. In datasets such as **Pick-a-Pic v2**, the preferred and unpreferred images are generated by different models (e.g., variants from the Stable Diffusion family, Dreamlike Photoreal 2.0) and then ranked by human annotators. These model-specific characteristics lead to diversity in outputs even when conditioned on the same prompt $c$, resulting in naturally divergent distributions between $y_w$  and $y_l$.
>
> Inspired by analogous observations in language models (e.g., \[3\]), we conducted a comprehensive empirical analysis of semantic similarity within the **Pick-a-Pic v2** dataset. As shown in **Figure 3** and **Figure 9**, the semantic similarity (measured by CLIPScore and VQAScore) between the preferred images and their prompts is often significantly higher than that of the unpreferred images. This supports our hypothesis that preference signal is reinforced by semantic misalignment.
>
> Following these insights, we generate semantically relevant captions and train our generative model using the **DCPO** loss function. Our experiments demonstrate that this approach leads to improved performance over existing methods across several benchmark datasets.
>
> Finally, to clarify the methodology in **Figure 3**: we evaluate the CLIPScore of image–prompt pairs ($y_w$, $c$) and ($y_l$, $c$) across the entire dataset. The prompts $c$ are shared across comparisons, and both $y_w$ and $y_l$ are sampled independently from different generative models conditioned on the same prompt. We will revise the figure caption and main text to ensure this setup is explicitly stated.
>
> \[1\] https://arxiv.org/pdf/2502.14560v2
> \[2\] https://aclanthology.org/2025.findings-acl.282.pdf
> \[3\] https://arxiv.org/abs/2404.00530
>
> > It is not well explained how Equation (8) is derived or verified. Specifically, how the use of two distinct captions (z_w, z_l) directly improves the optimization process. How exactly does this modification alter the RLHF-style objective?
>
> Thank you for your comment. As noted in Appendix A.1, we derive DCPO as a principled extension of the Diffusion-DPO formulation. Furthermore, in Appendix A.2, we have provided a series of theoretical analyses and proofs demonstrating the effectiveness of DCPO compared to the original Diffusion-DPO. These results support the soundness and practical advantages of our proposed approach.

---

### Decision · Action_Editor_Dzgj · 2025-09-12

**Recommendation:** Accept with minor revision

**Additional Comments:**

All reviewers recognize the merits of empirical advantages over baseline methods like Diffusion-DPO, MaPO, and Diffusion-KTO. These results, along with hyperparameter ablations of baseline methods and comparisons of different perturbation strategies, validated the core claims of mitigating irrelevant prompts.

The paper offers an interesting observation with strong empirical results. While two reviewers comment that the intuition is not fully grounded (i.e., the conflicting distributions), the AE believes that it's not a valid reason for rejection. The AE thus recommends to accept with a minor revision.

In the revision, the authors are strongly encouraged to revise the paper that
1) Include the additional experimental validations and comparisons in the rebuttal to the main paper
2) Justify and clarify the notion of conflicting distributions or tune down such motivations and present the method as a form of data augmentation with stronger performance.

**Audience:**

Yes

**Audience Explanation:**

Preference optimization for text-to-image diffusion models is of interest to the TMLR community.

**Claims And Evidence:**

Yes

**Claims Explanation:**

The paper proposes Dual Caption Preference Optimization (DCPO), which uses distinct captions for preferred and less preferred images to resolve conflict distribution and irrelevant prompt issues in preference optimization for text-to-image diffusion models, leading to improved performance gains over prior methods.

The performance improvement has been validated through extensive experiments across multiple baselines, metrics, and ablation study.